# An approximate stochastic optimal control framework to simulate nonlinear neuro-musculoskeletal models in the presence of noise

**Tom Van Wouwe** [1] *, **Lena H. Ting** [2,3], **Friedl De Groote** [1]

**1** Department of Movement Sciences, KU Leuven, Leuven, Belgium, **2** W.H. Coulter Department of Biomedical Engineering, Emory University and Georgia Institute of Technology, Atlanta, Georgia, United States of America, **3** Department of Rehabilitation Medicine, Division of Physical Therapy, Emory University, Atlanta, Georgia, United States of America

\* tvwouwe@stanford.edu

**Data Availability Statement:** No original experimental data was produced. Code is available on GitHub at https://github.com/

## Abstract

Optimal control simulations have shown that both musculoskeletal dynamics and physiological noise are important determinants of movement. However, due to the limited efficiency of available computational tools, deterministic simulations of movement focus on accurately modelling the musculoskeletal system while neglecting physiological noise, and stochastic simulations account for noise while simplifying the dynamics. We took advantage of recent approaches where stochastic optimal control problems are approximated using deterministic optimal control problems, which can be solved efficiently using direct collocation. We were thus able to extend predictions of stochastic optimal control as a theory of motor coordination to include muscle coordination and movement patterns emerging from non-linear musculoskeletal dynamics. In stochastic optimal control simulations of human standing balance, we demonstrated that the inclusion of muscle dynamics can predict muscle co-contraction as minimal effort strategy that complements sensorimotor feedback control in the presence of sensory noise. In simulations of reaching, we demonstrated that nonlinear multi-segment musculoskeletal dynamics enables complex perturbed and unperturbed reach trajectories under a variety of task conditions to be predicted. In both behaviors, we demonstrated how interactions between task constraint, sensory noise, and the intrinsic properties of muscle influence optimal muscle coordination patterns, including muscle co-contraction, and the resulting movement trajectories. Our approach enables a true minimum effort solution to be identified as task constraints, such as movement accuracy, can be explicitly imposed, rather than being approximated using penalty terms in the cost function. Our approximate stochastic optimal control framework predicts complex features, not captured by previous simulation approaches, providing a generalizable and valuable tool to study how musculoskeletal dynamics and physiological noise may alter neural control of movement in both healthy and pathological movements.

tomvanwouwe1992/SOC_Paper Code runs in MATLAB R2020a.

**Funding:** We gratefully acknowledge FWO (Research Foundation, Flanders: https://www.fwo.be/en/) for the fellowships FWO 1S82320N funding TVW and G079216N funding FDG that allowed to perform this research. The funders had no role in study design, data collection and analysis, decision to publish, or preparation of the manuscript.

**Competing interests:** The authors have declared that no competing interests exist.

## Author summary

Model-based simulations have advanced our insight in how movement is controlled, but computational limitations have prevented us from simultaneously considering the effects of nonlinear musculoskeletal dynamics and noise on neural control on movement strategies. Here we present a novel simulation framework that addresses this methodological gap and demonstrate its potential to predict features of motor control that have not been predicted previously. Our framework enables neural control mechanisms to be simulated and the optimal control strategy in the presence of noise to be computed. We demonstrate for the first time that muscle co-contraction–a motor strategy thought to be energetically costly–can help to reduce the effort required to stabilize standing postural control, even when feedback is present. We also demonstrate that multi-segmental models of the arm actuated by muscle can predict complex reach and perturbed reach trajectories that were not explained by simplified models. The ability to predict muscle activation patterns and kinematics of joints extends the theory of optimal control to variables that can be used to directly explain muscle and joint coordination. Our computational approach is broadly applicable and may be extendable to other normal and impaired movements, as well as aid in the design of assistive devices.

## Introduction

Predictive simulations are powerful tools to study the neuromechanics of movement [1,2]. Movement simulations are typically based on optimal control as a theory of motor coordination to solve the redundancy problem, i.e. to determine which of the many possible movement strategies is used to achieve a given movement goal [3]. Optimal control theory has also been used to demonstrate how sensorimotor noise affects movement control strategies. However, current approaches focus either on detailed representations of musculoskeletal dynamics while neglecting physiological noise [4–6] or on simulating the effects of physiological noise while simplifying musculoskeletal dynamics [7–9]. On the one hand, deterministic simulations based on complex models have advanced our understanding of how musculoskeletal dynamics shapes movement. On the other hand, stochastic simulations based on simple models have shown that noise shapes movement kinematics and variability as well as underlying control mechanisms, including modulation of feedback. Accounting for the interaction between musculoskeletal dynamics and physiological noise may therefore be important to predict physiologically-realistic control strategies and movements. However, the limited efficiency of available computational tools to simulate motor behavior in the presence of noise has hampered the use of more accurate models of musculoskeletal dynamics in stochastic simulations of movement [10]. Here, we present and test a generalizable computational framework to simulate the effect of sensorimotor noise on motor control and movement in nonlinear musculoskeletal systems.

Accounting for sensorimotor noise in movement simulations is crucial to capture movement variability and sensorimotor feedback modulation. In 1998, Harris and Wolpert minimized endpoint variability in the presence of signal-dependent motor noise to simulate open-loop controlled reaching and saccadic eye movements [11]. Including signal-dependent motor noise led to physiologically realistic bell-shaped velocity profiles and the experimentally observed trade-off between reaching time and reaching accuracy, i.e., Fitts' law. Although Harris and Wolpert clearly demonstrated the importance of accounting for noise, their stochastic simulations did not include feedback control. In general, feedback control improves

performance over open-loop control in the presence of sensorimotor noise [12]. Therefore, Todorov and Jordan introduced feedback control in stochastic simulations of movement and tested optimal feedback control (OFC) as a theory of motor coordination [3,13]. The most important prediction of optimal feedback control is arguably the minimum intervention principle: "deviations from the average trajectory are only corrected when they interfere with the task goal" [13]. Optimal feedback control has since explained many kinematic and control features of reaching and standing balance [14–17]. However, these studies mostly used linearized models that do not account for critical nonlinearities in movement.

Some studies have accounted for nonlinearities in musculoskeletal dynamics but relied on limiting assumptions on the control policy. For example, Li and Todorov [18–20] introduced iterative linear-quadratic-Gaussian control (iLQG), which solves the nonlinear stochastic optimal control problem by iteratively updating feedforward and feedback controls based on a local linearization of the dynamics around the current state trajectory. However, iLQG requires the control law to be time-varying, which might not be physiologically realistic. In addition, it requires a quadratic cost function and it requires specifying movement accuracy by a penalty term in the cost function [18,21]. As a result, the weights in the cost function need to be hand-tuned to achieve realistic results and an obligatory trade-off between accuracy and effort that may not necessarily be physiological emerges. Stochastic optimal control simulations of movement based on nonlinear models have remained rare, possibly because of their high computational cost.

Recent computational advances have drastically improved the efficiency of deterministic movement simulations enabling the use of complex and nonlinear musculoskeletal models and their application to stochastic movement simulations opens perspectives to further improve the realism of movement simulations. The introduction of direct collocation and automatic differentiation greatly speeded up simulating deterministic movements enabling the use of complex and nonlinear musculoskeletal models. In deterministic simulations, optimal control can be described by open-loop control trajectories. Therefore, these simulations can be formulated as trajectory optimization problems [5]. The introduction of direct collocation approaches to solve trajectory optimization problems improved computational efficiency compared to shooting methods by decreasing the sensitivity of the optimization objective to the decision variables [22]. Shooting methods use time-marching integration to evaluate the cost and constraint functions based on the current guess of the initial state and controls. Due to time-marching integration over the entire movement horizon, shooting methods have poor convergence and long computation times when the dynamics are stiff, as in many biological movements. In contrast to direct shooting, direct collocation eliminates the need for time-marching integration by adding the parameterized states to the decision variables and by adding the discretized dynamic equations as constraints to the optimization problem. Computational efficiency has further been improved by implicit formulations of the system dynamics to improve the numerical condition of the optimization problem [23] and the use of automatic differentiation to compute derivative information needed by gradient-based solvers [24]. These methodological advances have enabled rapid predictive simulations based on complex musculoskeletal models [2,6], which have been applied to test optimality principles underlying human movement [4] and to study the effect of changes in the musculoskeletal system on movement [6,25].

Computational advances that have improved the efficiency of deterministic simulations have been leveraged to stochastic optimal control simulations by reformulating stochastic simulations as approximate deterministic simulations. This approximate reformulation was originally applied for control in robotics and engineering [26]. Houska et al. [26,27] and Gillis et al. [28] proposed to transform the stochastic optimal control problem into an approximate

augmented deterministic optimal control problem by approximating the generally non-Gaussian state trajectory distribution by a Gaussian state trajectory that can be described by the mean state trajectory and the state covariance trajectory. The propagation of the covariance matrix is described by Lyaponuv differential equations, which assume local invariance of the system dynamics around the mean trajectory similar to the Extended Kalman Filter [29]. Berret et al. have applied this approach to simulations of reaching [30]. However, they only considered feedforward control [30] or added feedback control to the pre-computed optimal feedforward control in a post-processing step using the LQG framework [31]. This does not seem to yield an optimal solution as during the optimization of the feedforward control policy, the presence of feedback is neglected. It is therefore likely that co-optimizing feedforward and feedback contributions might result in better and more realistic control policies.

Here, we apply the approach based on approximating stochastic simulations as deterministic simulations to simultaneously compute feedforward and feedback contributions to control of nonlinear musculoskeletal dynamics in the presence of sensorimotor noise. We present a general formulation of the approach that is applicable to a broad range of movements, described by nonlinear dynamics, corrupted by additive and/or signal-dependent Gaussian noise, and controlled by time-varying feedback laws with any temporal and structural design, any type of cost function. Finally, our approach allows formulating movement accuracy as a task constraint, which may be more representative of the actual task-level goal.

We first show how our stochastic optimal control framework enables the prediction of interactions between muscle co-contraction and sensorimotor feedback, using human standing balance control as an example. While OFC has been used to predict sensorimotor feedback control of balance, individual muscles and muscle dynamics have not been modeled and it is therefore unclear whether OFC also predicts muscle co-contraction as a complementary feedforward control strategy. Specifically, OFC simulations based on joint torque-driven mechanical models capture modulation of feedback control with changes in sensory acuity [32,33], i.e. sensory reweighting. We hypothesized that OFC simulations will also capture modulation of feedforward co-contraction when history dependent muscle properties such as short-range-stiffness (SRS) [34–36] are taken into account. We simulated perturbed (platform rotations and translations) standing balance based on a multi-sensory, muscle-driven model with both feedforward and feedback control in the presence of sensory and motor noise. We demonstrate that our stochastic optimal control framework predicts contributions of both feedforward, i.e. muscle co-contraction, and feedback control during standing balance that depend on movement task, sensory acuity and muscle properties.

We next demonstrate that the stochastic optimal control framework can predict perturbed reach trajectories as well as the underlying control policy consisting of feedforward and feedback contributions when nonlinear musculoskeletal dynamics are considered. Prior OFC simulations based on a point-mass model capture changes in nominal reach trajectories and feedback control depending on target shape and the presence of obstacles (i.e. task goal) [8]. However, perturbed reach trajectories deviated considerably from experimental observations. We hypothesized that more accurate representations of nonlinear multi-joint and muscle mechanics would result in more realistic reach trajectories. We simulated reaching while altering target shape and stability of the environment (i.e. divergent force-field) [37,38]. We demonstrate that our stochastic optimal control framework using a muscle-driven arm model results in improved predictions of reach kinematics in perturbed conditions as well as detailed predictions of sensorimotor feedback and muscle level control. In addition, our model predicted muscle activity in response to perturbations, which was not possible with previous simulation models, in agreement with experimental data [8].

## Results

### Approximate stochastic optimal control framework

We simulated movement trajectories and movement variability based on nonlinear musculo-skeletal dynamics driven by optimal feedforward and feedback control policies in the presence of sensory and motor noise. For each specified task, optimal control policies were computed by minimizing the expected effort in the presence of noise. Effort was defined as the time integral of the sum of muscle excitations squared [39], leading to the following general stochastic optimal control problem:

$$\min_{e_{ff}(t),K(t)} \quad : \quad J = E[\int_{t_{start}}^{t_{final}} e^{T}(t)e(t)dt]$$

$$\text{subject to} : \quad \dot{x}(t) = f(x(t), e(t), w_m),$$

$$g(x(t), e(t)) \geq 0$$

$$e(t) = e_{ff}(t) + K(t) \cdot y_{fb}(x(t), w_s)$$

with $E[]$ the expected value function, $x(t)$ the stochastic state trajectory, including joint kinematics and muscle activations, $e(t)$ the stochastic muscle excitation trajectories, $w_m$ a set of zero-mean Gaussian motor noise sources. The musculoskeletal dynamics $f(x(t),e(t),w_m)$ were stochastic and nonlinear. To specify different task goals, we imposed task-dependent path constraints and bounds $g(x(t),e(t))$. Muscle excitations $e(t)$ consisted of time varying deterministic feedforward muscle excitations, $e_{ff}(t)$, as well as feedback muscle excitations derived from a linear feedback law with deterministic time-varying feedback gains $K(t)$ and task-dependent feedback error signals $y_{fb}(x(t), w_s)$. We did not separately model short- and long-latency reflexes and considered task-level sensory feedback rather than local sensory feedback. Here, we modeled sensory noise $w_s$ as zero-mean Gaussian noise added to the feedback error signal $y_{fb}$. These stochastic optimal control problems were approximated by deterministic optimal control problems and then solved using direct collocation (for details, see Methods). The deterministic approximation was based on the assumption that the stochastic state trajectories could be modelled by a Gaussian distribution and could thus be described by their expected value (mean trajectory) and variance (state covariance matrix).

### Contributions of muscle co-contraction and feedback control in eyes-closed perturbed standing balance depend on movement task, sensory acuity and muscle properties

We first demonstrate that stochastic optimal control can simultaneously predict muscle co-contraction and sensorimotor feedback contributions to motor coordination, using eyes-closed perturbed standing balance as an example. Although OFC can capture experimentally identified modulations of feedback contributions that depend on sensory acuity and perturbation type [33,40–42], it is unclear whether OFC predicts experimentally-observed muscle co-contraction complementing feedback during perturbed standing balance. To address limitations of prior models, we simulated standing balance with eyes closed using an inverted pendulum model of the body that was driven by a pair of antagonistic ankle muscles that have activation-dependent impedance (Fig 1A). We further investigated how the predicted muscle co-contraction depends on the model of muscle mechanical impedance, performing

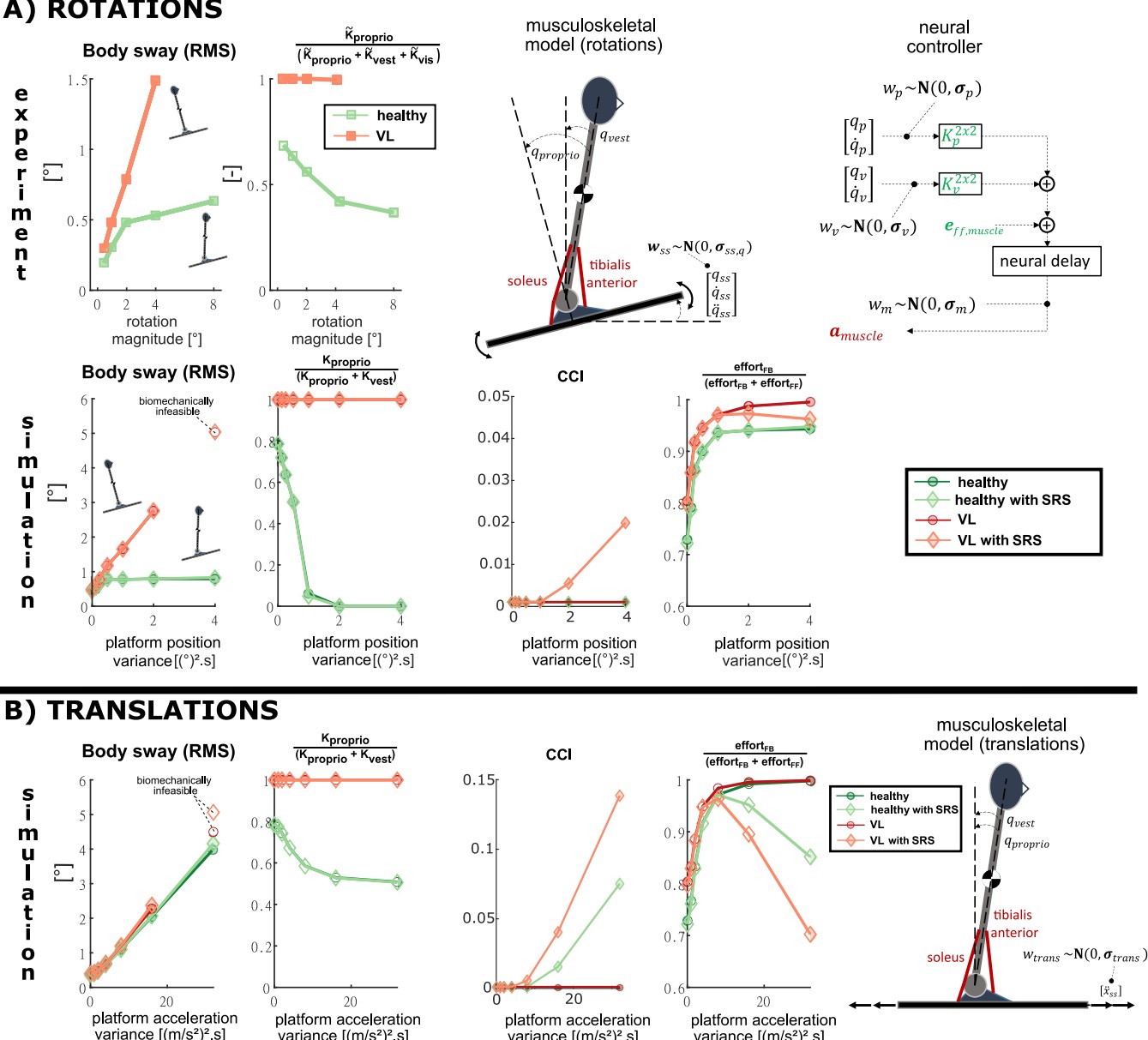

**Fig 1. A) ROTATIONS. Musculoskeletal and control model.** The proprioceptive (relative angle between the body and platform: $q_p, \dot{q}_p$) and vestibular (absolute body angle: $q_v, \dot{q}_v$) cues encode different information that is corrupted by proprioceptive ($w_p$) and vestibular noise ($w_v$). The platform rotations are modeled by Gaussian noise ($w_{SS}$) of the platform position ($q_{SS}$), velocity ($\dot{q}_{SS}$) and acceleration ($\ddot{q}_{SS}$). $a_{muscle}$ are muscle activations; $e_{ff,muscle}$ are muscle feedforward excitations that are constant here; $K_p^{2x2}$ are proprioceptive feedback gains and $K_v^{2x2}$ are vestibular feedback gains. **Experiment and simulation.** Experimental data for body sway and feedback gains are from Peterka et al. [32]. RMS body sway, relative experimental proprioceptive feedback gains $\frac{\|\tilde{K}_{proprio}\|}{\|\tilde{K}_{proprio}\| + \|\tilde{K}_{vest}\| + \|\tilde{K}_{vis}\|}$ and relative simulated proprioceptive feedback gains $\frac{\|K_{proprio}\|}{\|K_{proprio}\| + \|K_{vest}\|}$, co-contraction index (CCI) and contribution of expected effort from feedback to the total expected effort: $\frac{effort_{FB}}{(effort_{FF} + effort_{FB})}$ for healthy and vestibular loss (VL) subjects with and without short-range-stiffness (SRS) modeled. **B) TRANSLATIONS. Musculoskeletal model.** The control model is identical to the model used to simulate the response to rotation perturbations, but in the case of translation perturbations the relative and absolute body angles are identical and vestibular and proprioceptive cues encode the same information. Platform translations are modeled by Gaussian noise ($w_{trans}$) of the linear platform acceleration ($\ddot{x}_{SS}$). Simulation results. **Simulation.** Same outcome variables as for the rotation perturbations are shown.

simulations using both a Hill-type muscle model that accounts for the force-length and force-velocity properties of muscles [22], and an augmented Hill-type muscle that also accounts for muscle short-range stiffness that was proportional to muscle activation (see Methods) [43,44]. To model baseline muscle activity during quiet standing, feedforward muscle excitations were modeled as constants. As in prior studies using torque-driven models of perturbed standing balance [15,32,45], feedback contributions were modelled as a linear combination of delayed (first order approximation with time constant of 150ms, see Methods) proprioceptive and vestibular cues, which encode the angle and angular velocity between the body and the platform, and gravity, respectively (Fig 1A). Since we simulated the eyes-closed condition our model assumes that no visual information is used for control. Gaussian sensory and motor noise was added to feedback signals and muscle excitations respectively. In accordance with the literature, vestibular inputs were more noisy than proprioceptive inputs [46]. We specified the task of upright standing by imposing a constant mean upright posture and a postural sway within limits of stability. To elicit a range of feedforward and feedback control policies, we simulated both random sagittal platform rotation (Fig 1A) and translation (Fig 1B) perturbations of different magnitudes. Co-contraction was quantified by the co-contraction index (CCI) introduced by Rudolph et al. [47]. Comparing solutions for platform rotations and translations allows sensory feedback and muscle co-contraction contributions to balance control to be differentiated as rotation and translation perturbations require different ankle muscle coordination to maintain equilibrium [48] (Fig 1A and 1B).

For platform rotations, stochastic optimal control predicted changes in postural sway and sensory reweighting with perturbation magnitude in agreement with experimental observations in healthy individuals and vestibular loss subjects (Fig 1A–simulations & experiments) [32,49,50]. In agreement with experimental findings by Peterka for healthy adults [36] (Fig 1A–experiments, healthy) and prior OFC results, body sway in our simulations increased quasi-linearly with increasing amplitude of platform rotations and saturated at larger amplitudes where the body moved in anti-phase with the platform [32,51,52] (Fig 1A–simulations, healthy). The simulations predicted sensory reweighting similar to that observed experimentally [32,52] (Fig 1A–experiments, healthy), with increased reliance on vestibular feedback at higher rotation magnitudes (Fig 1A–simulations, healthy). Because proprioceptive information was modelled to be more accurate than vestibular information [48], there was a higher reliance on proprioception and the body moved in-phase with the platform at low rotation magnitudes. When removing vestibular sensory information to simulate vestibular loss subjects, the simulation predicted a quasi-linear increase in sway with platform rotation amplitude (Fig 1A–simulations, vestibular loss) consistent with experimental observations [52] (Fig 1A–experiments, vestibular loss). In the case of vestibular loss, the relative weighting of proprioceptive information was 100% in simulation (Fig 1A–simulations, vestibular loss) and nearly 100% in experiments (Fig 1A–experiments, vestibular loss) throughout platform rotations. Peterka et al. [52] determined sensory weighting by identification of the feedback gains $\tilde{K}_{proprio}, \tilde{K}_{vest}, \tilde{K}_{vis}$ of a linear model based on collected sway data, with $\tilde{K}_{vest} = 0$ for vestibular loss subjects. In this study [52], the identified $\tilde{K}_{vis}$ in the eyes-closed condition were close to but not exactly 0, explaining the small deviations from 100% proprioceptive weighting in this condition. In contrast to sway for healthy subjects, sway for vestibular loss subjects was predicted to follow the platform motion, as proprioception is the only source of sensory information [53]. The model predicted higher total effort in vestibular loss subjects compared to healthy subjects and loss of balance when simulated sway amplitudes became unrealistic (RMS sway values larger than 5˚).

During platform rotations, slightly higher, but still low levels of co-contraction were predicted for vestibular loss subjects [54] than for healthy controls (Fig 1A–simulations, CCI in healthy with SRS and VL, model with short-range stiffness). In rotations, increased joint impedance due to co-contraction opposes the anti-phase movement of the ankle joint that is optimal for upright balance with minimal effort. However, co-contraction contributes to the strategy of maintaining a constant joint angle with respect to the platform, the strategy predicted when the reference to gravity is absent, as in vestibular loss subjects [32,49,55]. Our simulations predicted co-contraction to increase above a certain perturbation magnitude in the absence of vestibular sensory information (Fig 1A–simulations, CCI in VL, model with short-range stiffness). Since all simulated strategies minimize effort, our simulations predict muscles co-contraction to reduce effort with respect to using feedback only at high perturbation magnitudes in vestibular loss subjects. However, as feedback control also increases in the absence of vestibular information, the proportion of effort due to feedback control is nevertheless higher in vestibular loss than in healthy subjects (Fig 1A–simulations, effort in VL, model with short-range stiffness).

In contrast to rotations, during platform translations stochastic optimal control predicted similar increases in postural sway with increasing perturbation magnitude in healthy and vestibular loss simulations, also found experimentally [56] (Fig 1B—simulations). As perturbation magnitude increased, the proportion of proprioceptive feedback used in balance control decreased, but in contrast to rotations, shifted toward an equal contribution of proprioceptive and vestibular feedback. In translations¸ both sensory signals encode the same information but with different uncertainty levels, therefore this shift towards more equal contributions with increasing perturbation magnitude may be explained by the sensory uncertainty becoming increasingly small in comparison to the kinematic deviations introduced by the perturbations. The decreased contribution of proprioceptive feedback with increasing translation magnitude has not been tested experimentally and should be considered a model prediction, allowing further validation of our simulations.

In contrast to rotations, and consistent with experimental findings [57], much larger contributions of muscle co-contraction were predicted with increasing translation perturbation magnitude in both healthy and vestibular loss subjects (Fig 1A and 1B–simulations, CCI). In translations, increased joint impedance due to co-contraction reduced body sway, which in turn reduced the relative effort of feedback corrections as perturbation magnitude increased (Fig 1B–simulations, effort).

Taken together, stochastic optimal control predicted muscle co-contraction as a complementary strategy to sensorimotor feedback depending on the mechanical properties of the muscle, perturbation type and magnitude, and sensory acuity. Muscle co-contraction was only predicted in the simulations where the Hill-type muscle model was augmented with short-range-stiffness [43]. Although some amount of muscle co-contraction was predicted in both rotation and translation perturbations, muscle co-contraction only considerably decreased the proportion of muscle effort due to feedback control in high-amplitude translation perturbations, where muscle co-contraction helps to maintain upright posture.

## Stochastic optimal control of goal-directed reaching predicts experimental kinematics, movement variability and feedback modulation across different reaching tasks

In a second set of simulations, we demonstrate that stochastic optimal control using a nonlinear muscle-driven arm model can predict how nominal as well as perturbed reaching trajectories change as a function of task goals and environmental dynamics. We simulated the three point-to-point reaching tasks (Fig 2A) described in Nashed et al. [8]: reaching to a circular

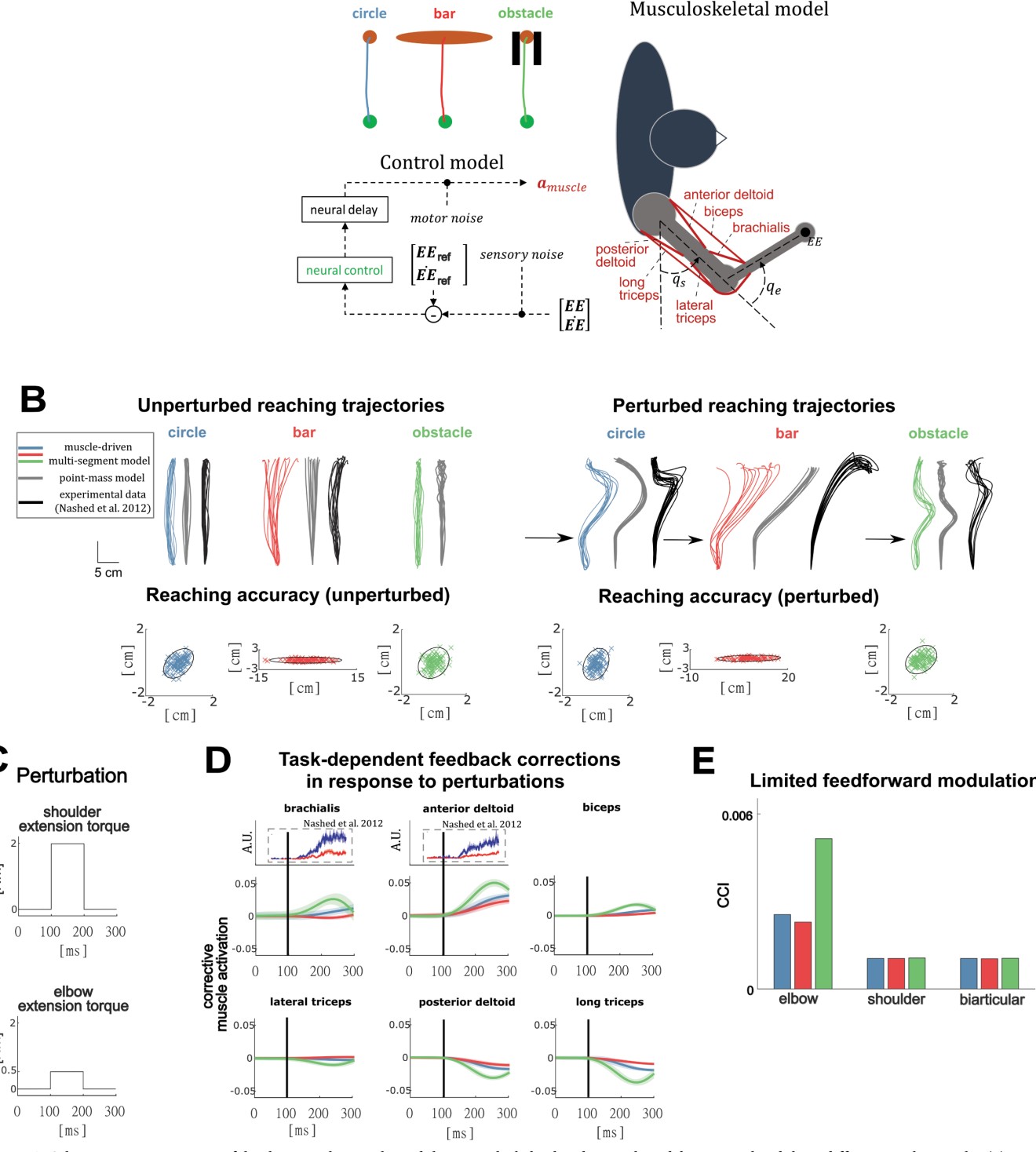

**Fig 2. A:** Schematic representation of the three reaching tasks and the musculoskeletal and control model. We simulated three different reaching tasks: (1) reaching towards a small circular target (blue), (2) reaching towards a horizontal bar (red), and (3) reaching towards a circular target in the presence of an obstacle (green). In all simulations, muscle excitations consist of feedforward and feedback contributions. The feedback controller is driven by the error between the end-effector kinematics ($EE$, $\dot{EE}$) and the nominal end-effector kinematics ($EE_{ref}$, $\dot{EE}_{ref}$) and is corrupted by sensory noise. Muscle forces are dependent on muscle activations ($a_{muscle}$), which are time-delayed muscle excitations, and resulting joint torques are corrupted by motor noise. **B:** Unperturbed

and perturbed reaching trajectories predicted by our model (colored), predicted with a point-mass model (grey) and measured (black). End-point accuracy simulated with our model for the three different reaching tasks. Ellipses denote 95% confidence regions. **C:** Perturbations applied during the perturbed reaching movements. **D:** Simulated muscle-level corrective actions that result from optimal feedback in response to unexpected extension perturbations. For elbow and shoulder flexor muscles (anterior deltoid, biceps) we show experimental EMG data from Nashed et al. [9]. **E:** Average simulated muscle co-contraction level for the shoulder and elbow joints for the three reaching tasks.

target (circle), reaching to a horizontal bar (bar), and reaching to a circular target in the presence of a narrow obstacle (obstacle) (Fig 2A). Previously these reaching tasks were simulated using a point-mass model of the arm assuming OFC. To capture nonlinear inter-segment dynamics, we modeled the arm as a planar two-segment kinematic chain. To enable muscle activity predictions, the arm model was driven by six Hill-type muscles with rigid tendons consisting of a uni-articular agonist-antagonist pair for the shoulder (posterior and anterior deltoid) and the elbow (brachialis and lateral triceps), and a bi-articular agonist-antagonist pair spanning both the elbow and the shoulder (biceps and long triceps) [20]. Net muscle excitations were a sum of time-varying feedforward excitations and delayed (first order approximation with time constant of 150ms, see Methods) time-varying linear feedback of the end effector (hand) kinematic error [8,10]. Gaussian sensory noise and motor noise were added to the feedback signals and joint torques, respectively. The three different tasks were modeled by constraining the end effector position variability to achieve the task requirements. Optimal feedforward controls, feedback gains and reference end effector trajectories were computed by minimizing expected effort during unperturbed reaching, and the resulting optimal control policies were then used to generate stochastic forward simulations of both unperturbed and perturbed reaching. Perturbations in the perturbed reaching simulations were modelled as external torques inducing shoulder and elbow extension (Fig 2C).

Like predictions based on the point-mass model, predictions based on our muscle-driven, multi-segment arm model were in line with the minimum intervention principle [3,58] where kinematic deviations are only corrected when they interfere with the task goal: horizontal deviations are left uncorrected in the bar condition. All our optimal control policies met the required reaching end-point accuracy, for both perturbed and unperturbed reaching, as imposed for the different tasks in the stochastic optimal control problem formulations and in accordance with experimental data [8] (Fig 2B, reaching accuracy).

In contrast to point mass simulations, our non-linear muscle-driven model predicted both unperturbed and perturbed reaching trajectories that were more similar to experimental findings [8]. In unperturbed reaches, an important improvement over predictions from a point mass model are the slightly curved mean trajectories in the circle and bar task, as have been experimentally observed [11] (Fig 2B, compare colored with black and grey trajectories). In the obstacle task the reach trajectories are similar for the point mass and non-linear muscle driven simulations. Similar to the experiments, the non-linear muscle driven simulations show less curvature for the obstacle task compared to the circle task as the second part of the motion the hand is constrained to be on a straight vertical line in the simulation.

Perturbed reach trajectories using our multi-segment muscle-driven arm model were in better agreement with experimental observations than perturbed reach trajectories using a point mass model. In agreement with experiments, and in contrast to point mass models, our model predicted late and steep corrections in the kinematic trajectory for the circle task (Fig 2B colored lines). Our model similarly predicted later corrections occurring over a shorter reaching distance than the point mass model for the obstacle task. Yet, these corrections happened sooner than experimentally observed. For the bar task our model did not predict the overshoot of the hand in the vertical direction that was observed in experiments (Fig 2B perturbed trajectories, red and black lines). The early, symmetric corrections predicted by the

point-mass simulations are likely due to the decoupled control of the vertical and horizontal degree of freedom. Given the similarity in the underlying control hypothesis between our simulations and the point-mass simulations of Nashed et al. [8], the improved agreement with experimentally observed reach trajectories indicates the importance of accounting for nonlinear dynamics when predicting reaching movements [58,59].

A novel aspect of our model is that muscle activations in response to perturbations could also be predicted and compared to available EMG recordings. Simulated corrective muscle activations were similar to recordings in the anterior deltoid and brachialis, with larger corrections in the 'circle' than in the 'bar' condition (Fig 2D, blue vs red). Simulated corrective muscle activations in the 'obstacle' (Fig 2D, green) condition were larger and peaked earlier than those in the circle and bar conditions. As EMG data was only available for the anterior deltoid and brachialis in the circle and bar conditions, simulated muscle activity of the antagonistic muscles and in the bar condition remains to be validated experimentally.

We found only limited muscle co-contraction with an average co-contraction index that was always below 0.006 (where 0 indicates no co-contraction and 2 indicates maximal co-contraction). The mean muscle co-contraction index was similar across task conditions for the uni-articular shoulder muscles and bi-articular muscles (Fig 2E). For the uni-articular elbow muscles co-contraction increased in the obstacle task. This was observed experimentally as well [8]. Note that we did not include short-range stiffness in our muscle model, which was necessary in the balance simulations to predict co-contraction, since short-range-stiffness contributions are negligible during movements over large ranges of motion [46]. The increase in co-contraction observed in the 'obstacle' condition will therefore have a small effect on joint impedance, suggesting that it follows from the altered feedforward trajectory needed to avoid the obstacle rather than from a shift towards feedback control to increase joint impedance.

## Stiffness is modulated during goal-directed reaching in a divergent force field through changes in feedback but not feedforward control

To explore how the dynamics of the environment influences predicted contributions of muscle co-contraction and feedback control, we simulated reaching to a circular target in the presence of a divergent force field (Fig 3). Although co-contraction has been observed during reaching in a force field, the relative contribution of muscle co-contraction and sensorimotor feedback control to increased endpoint stiffness is not known [38,60,61]. We computed end-point stiffness as is done experimentally [62], i.e., by perturbing the hand in a specific direction in simulation and by dividing the resulting change in endpoint force by endpoint displacement at the end of a short-time interval (150ms).

In agreement with experimental data [37,61], simulated reaching accuracies in a force field were similar to reaching accuracies in the absence of the force field (Fig 3A). Our multi-segment, muscle driven model predicted straighter reaching trajectories in a divergent force field than in the absence of a force field, which can be explained by the additional effort required to counteract the force field when not on a straight line (Fig 3A). Next, after a perturbation, corrections were performed more rapidly as the hand was moved sooner towards the nominal trajectory in the presence of a force field than in the absence of a force field. This strategy reduces effort compared to slower corrections in the presence of the force field.

Similar to the experiments performed by Burdet et al. [37] and Franklin et al. [38] our stochastic optimal control model predicted increased stiffness in the horizontal direction but not in the reaching direction in the presence of a horizontal divergent force field of 200 N/m (Fig

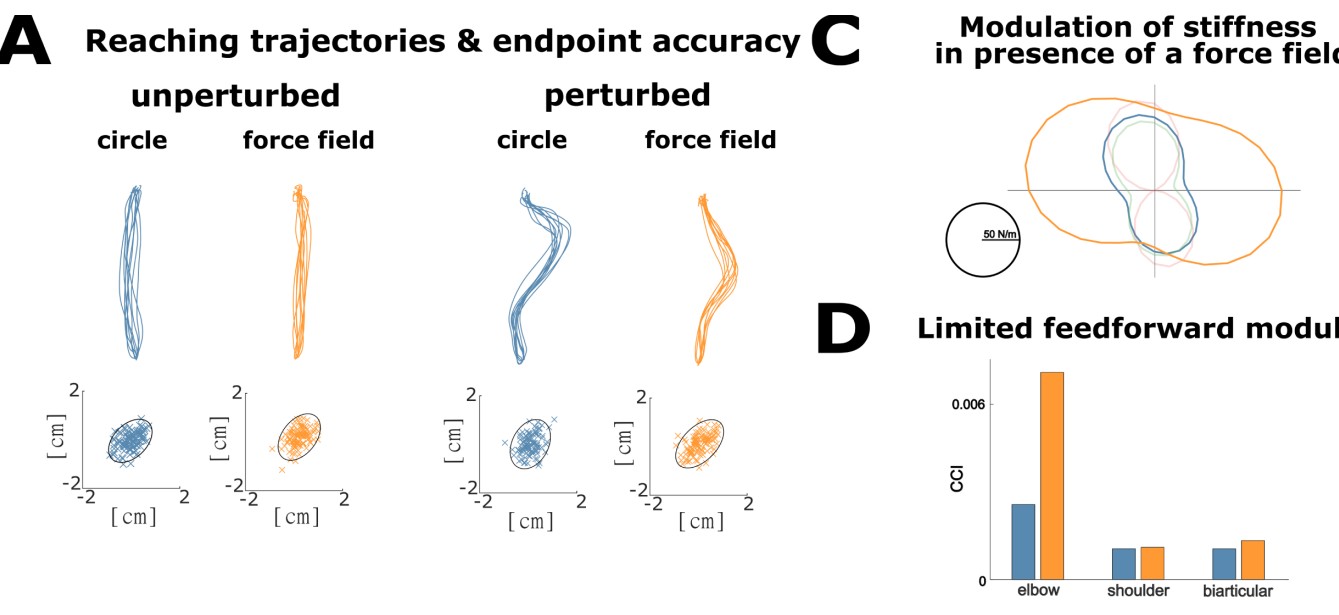

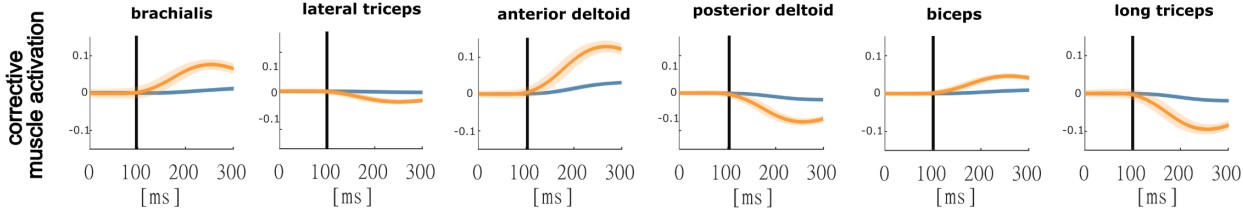

**Fig 3. A:** Unperturbed and perturbed reaching trajectories predicted in the presence ('force field') and absence ('circle') of a 200N/m divergent force field. Simulated end-point accuracy ellipses denote 95% confidence regions. **B:** Simulated muscle-level corrective actions that result from optimal feedback in response to unexpected extension perturbations. **C:** Horizontal and vertical end-effector stiffness throughout the reaching movement for the optimal controllers in the absence and presence of the divergent force field, and for the other reaching tasks (in absence of the force field). **D:** Average muscle co-contraction level for the uni-articular shoulder and elbow muscles and the bi-articular muscles in the presence and absence of the force field while reaching to the circle.

3C). The simulated orientation of the stiffness ellipses, with main axes tilted with respect to the horizontal and reaching directions, in both the presence and absence of a force field is in agreement with experimental observations [37,38] but differs from simulations based on a point-mass model, where the main axes of the stiffness ellipse are aligned with respect to the horizontal and reaching directions respectively [30].

In our model, the optimal strategy to increase stiffness in the presence of a divergent force field was to upregulate feedback control to all muscles without increasing co-contraction, while in experiments co-contraction increased significantly [38,58,60,61] (Fig 3B and 3D). The small increase in simulated co-contraction around the elbow in the divergent force field (Fig 3D) likely resulted from small changes in the mean kinematic trajectory and hence feedforward excitations when the force field was applied.

The simulated corrective activity of all muscles in response to perturbations was higher when reaching in a divergent force field than in a stable environment (Fig 3B). These predictions on corrective muscle activity in environments with different stability as well as perturbed reach trajectories in a divergent force field remain to be validated in future experiments.

## Discussion

Our major contribution was to apply a recently developed efficient approximate stochastic optimal control framework to simulations of motion enabling us to predict, for the first time, realistic movement trajectories and muscle coordination patterns emerging from nonlinear musculoskeletal dynamics, feedforward, and feedback neural control in the presence of noise. We used a generally applicable method to approximate the true stochastic optimal control problems by deterministic optimal control problems, which could be solved efficiently. The use of direct collocation and gradient-based optimization to solve the approximate deterministic problems facilitated the use of constraints, allowing us to distinguish effort optimization and task-level goals, such as accuracy, rather than to trade them off in the cost function. The framework allowed us to considerably extend the predictions of stochastic optimal control as a theory of motor coordination to relative contributions of feedforward and feedback control of non-linear musculoskeletal systems. The use of nonlinear mechanical models that captured multiple joints and muscles enabled detailed comparison to experimental kinematic and EMG data. In particular, the addition of muscle models in stochastic optimal control simulations, demonstrated that muscle co-contraction can—in the presence of uncertainty in sensory information—minimize muscle effort required for a task. In balance control simulations, we demonstrated that benefits of increased impedance from muscle co-contraction depend on the intrinsic properties of the muscle and the interaction with delayed sensorimotor feedback mechanisms. Similarly, in reaching simulations, we demonstrated that multi-segment musculoskeletal dynamics are key to predicting complex perturbed and unperturbed reach trajectories under different task conditions, resulting from interactions between feedforward and feedback sensorimotor control mechanisms. Taken together, our results showed that complex features of motor control and movement that were not captured by previous optimal control simulations can emerge from nonlinear stochastic optimal control processes. As such, our approximate stochastic optimal control framework provides a valuable tool to providing insight in neuromechanics of normal and possibly impaired movement that incorporates the complexities of the neuromusculoskeletal system and the effects of physiological noise.

Implementing a direct collocation approach to the solution of nonlinear stochastic optimal control problems increased computational efficiency and modelling flexibility, allowing neuro-musculoskeletal model complexity, and thereby fidelity, to be increased. Most prior methods have only allowed efficient solutions of linear, but not nonlinear, stochastic optimal control problems [7–9,63], and were therefore not capable of predicting many important movement features. Stochastic optimal control simulations based on nonlinear dynamics have previously been performed using iterative Linear-Quadratic-Gaussian (iLQG) methods for reaching tasks [18]. However, such methods require a time-marching integration of the system dynamics under the current guess of the control law at each iteration to evaluate the nominal controls and state trajectories. Such shooting methods might be less suitable than collocation methods for movement tasks with less stable system dynamics, such as standing or walking. Further, the LQG approach assumes a quadratic cost function and a linear and continuously time-varying feedback control law, which might not always represent the biological system. Finally, LQG methods require constraints on the mean state and state covariance (e.g. limiting the standard deviation of the end-point position of the hand in a reaching task) to be imposed through penalty terms in the cost function, whose weights require tedious tuning. In contrast, the approach based on collocation and gradient-based optimization we used here allows imposing such constraints in a direct and intuitive manner. A few previous studies [64,65] have relied on direct collocation methods as well to solve stochastic optimal control problems but they evaluated the stochastic cost function and path constraints based on the simulation of

a limited number of noisy episodes rather than describing the state distribution. For nonlinear systems with a limited number of degrees of freedom, such a sampling based approach appears tractable. However, with increasing degrees of freedom, the number of noisy episodes that is required to capture the underlying stochastic dynamics might become computationally intractable. Simulations based on too few episodes might result in problematic unstable and suboptimal solutions. We overcame the limitations of prior methods by applying a recently proposed method to approximate stochastic optimal control problems by deterministic optimal control problems [26,27]. The proposed approach is based on the assumption that the state distribution can be approximated by a Gaussian distribution and that the propagation of the state covariance can be described by the Lyapunov equation [66]. This framework is applicable to optimal control problem formulations with any cost function design while feedback laws with any temporal and structural design are possible. Here, we used it to simultaneously solve for feedforward and feedback control laws.

Our novel framework allows task-goals to be specified as constraints allowing us to evaluate the minimal effort solution for a given task. It is still debated to what extent optimality assumptions capture human movement and what the optimality criteria underlying human movement are [67]. Previous stochastic optimal control simulations have typically been based on a multi-objective cost function reflecting a trade off in effort and accuracy [3,11,33]. In such simulations, the weights in the cost function must be tuned to produce simulations that match the desired accuracy. By using constraints to impose accuracy requirements and other task goals, task requirements can be separated from optimality principles that govern task execution within these requirements. We reproduced key features of movement kinematics and control by using expected effort as the sole performance criterion. Not only does the approximate stochastic optimal control framework remove the need of building multi-objective cost functions, which might require considerable user intuition, minimizing effort within the solution space allowed by the task requirements might also be more representative of how humans approach a task. Rather than minimizing variability, humans might pick a control strategy that is 'good enough' [68]. For example, in standing a 'good enough' control strategy prevents a fall and for reaching a 'good enough' control strategy brings the hand in the target. The framework, based on direct collocation, can efficiently handle constraints offering flexibility in formulating the optimal control problem by for example imposing control bounds and constraining kinematic variability throughout movement execution in agreement with task requirements. Our approach thus allowed us to further test effort minimization as the optimality criterion underlying human movement.

Our stochastic optimal control framework predicted co-contraction as a minimal effort strategy during perturbed standing balance, suggesting that a combination of co-contraction and feedback corrections is energetically more efficient than feedback corrections only. Our simulations suggest that muscle co-contraction might be a minimal effort strategy to achieve movement goals whereas it has often been seen as an energetically costly strategy to maximize accuracy. Co-contraction reduces effort when task performance benefits from increased joint impedance and activation-dependent muscle properties allow to increase intrinsic mechanical impedance at reasonable costs. This seems to be the case in healthy subjects for translational but not rotational perturbations and in vestibular loss subjects for both translational and rotational perturbations but only when muscle short-range stiffness was taken into account. Prior models also predicted feedback modulation with changing sensory acuity, but did not include muscles that allowed for co-contraction as a complementary strategy [15,45]. Modeling assumptions might have influenced the simulated levels of co-contraction in standing balance control. We used an all or nothing approach to model short-range-stiffness whereas short-range-stiffness disappears when muscle stretch exceeds a certain threshold [44]. Therefore,

short-range-stiffness might only contribute during low amplitude sway up to 3° [35,44], and we might have overestimated its contribution when platform perturbations were high enough to induce large sway. Similarly, we might have overestimated contributions from co-contraction by modeling rigid tendon, especially when accounting for short-range stiffness (see also S1 Text).

Muscle co-contraction has been observed experimentally during reaching [69], especially in a divergent force field [58,60,61], but was not predicted by our stochastic optimal control simulations. Berret et al. only considered feedforward control in their simulations of reaching and found higher levels of co-contraction [30] as in the absence of feedback control, co-contraction is the only mechanism to increase joint impedance and hence robustness against perturbations during reaching. Our simulations suggest that muscle co-contraction is an energetically costly strategy when feedback is present resulting in very low simulated levels of co-contraction. Hence, both the simulations of Berret et al. [30] and our simulations are in conflict with the experimental observation of both co-contraction and feedback corrections during reaching. We might have failed to find co-contraction in our simulations because our framework does not allow predictions of the agonist-antagonist dual control strategy depending on muscle co-contraction [10]. By co-contracting antagonistic muscles, it is possible to increase agonist activity and decrease antagonist activity simultaneously yielding a more efficient response to a perturbation. Although our approach to stochastic optimal control accounts for nonlinear state dynamics, it was still based on a linear approximation around the mean state trajectory to propagate the state covariance matrix. An important nonlinearity arises from muscle activity being bound between zero and one and hence, reductions in muscle activity are only possible when muscle activity is larger than zero. Yet this nonlinearity is not reflected in our approximation of the state covariance dynamics, which resulted in predicted reductions in muscle excitations even when this would result in muscle excitations becoming smaller than zero. Hence, there was no need to increase baseline activity in our simulations to exploit the agonist-antagonist dual control strategy, which likely resulted in overly large contributions of this strategy as they came at no cost. To further improve the realism of our simulations, we thus need to more accurately describe the nonlinearities in the dynamics of the state covariance.

Several other modeling assumptions might have influenced the simulated levels of co-contraction. First, we did not model that co-contraction increases short-latency reflexes through gain scaling [70,71], thereby underestimating the contribution of co-contraction to reflexes. Second, due to the first order approximation to model feedback delays, sensory information–although attenuated—is available immediately. This might have increased the efficiency of feedback corrections, which in turn might have reduced the need for muscle co-contraction to increase joint impedance and provide instantaneous resistance against perturbations. Note, however, that the presence of sensorimotor noise prevented simulated feedback gains and therefore 'early' feedback contributions from becoming too large. Including an exact delay within the framework is important future work, and approximating delays by a low-pass filter must be done with caution when drawing conclusions. Third, feedback gains were unconstrained in how they could change as a function of time. Such infinite bandwidth is probably unrealistic and more realistic models of control dynamics might reduce the efficiency of feedback control and thereby increase simulated levels of co-contraction. Although the level of co-contraction might be sensitive to each of the above modeling assumptions, our simulations that simultaneously solved for optimal feedforward and feedback control contributions demonstrated that both muscle co-contraction and feedback gains varied as a function of the task, mechanical properties of the muscles, and sensory acuity.

A major contribution of our work was to create a framework for predicting how stochastic optimal control principles apply at the level of individual joints and muscles, which is difficult to impossible with currently available tools. Generating simulations that predict execution-level physiological variables that can be compared to experimental measures requires the subsystems, i.e. joints and muscles, from which those measures are obtained to be modeled. As such the ability of stochastic optimal control to predict movement cannot be accurately validated using simple models. While it is known that using different levels of detail to represent the musculoskeletal dynamics leads to different simulated responses to perturbations if motor commands remained unchanged [72], optimal control predictions can inform us on how the nervous system can exploit the nonlinear dynamics of the musculoskeletal system. For example, our two-segment, muscle-driven arm model yielded more realistic reach trajectories when reaching to different targets, when compared to prior predictions based on a point mass [8]. We further explored the sensitivity of predicted reach trajectories to the presence of bi-articular muscles and muscle properties (see S1 Text for details). Interestingly, simulated reach trajectories were robust against the removal of bi-articular muscles but sensitive to how monoarticular muscles were modeled. Using an alternative, physiologically plausible set of muscle properties yielded perturbed reach trajectories in the 'bar' condition that captured the experimentally observed overshoot. However, the overshoot was also predicted for perturbed reach trajectories in the 'circle' condition, where it was not observed experimentally. These exploratory results demonstrate both the potential of our framework to explore the effect of musculoskeletal properties on (perturbed) movement and the importance of accurate musculoskeletal models.

Simulations of endpoint stiffness in an unstable force field show, similar to experimental observations [58,60], that stiffness is modulated depending on the environment and that the orientation of the main axes of the stiffness ellipses are tilted with respect to the reaching direction as a result of musculoskeletal dynamics. Berret et al. [31] combined a similar musculoskeletal model with a different control strategy, i.e. feedforward control in the absence of feedback control, and found a similar tilt of the stiffness ellipses, suggesting that musculoskeletal dynamics is a determining factor. These examples demonstrate how stochastic optimal control applied to muscle-driven models can be used to interpret complex motor coordination at the muscle level and to study control of redundant sets of muscles in the presence of noise.

Leveraging computational advances from deterministic movement simulations to enable predictions for stochastic movement simulations may yield much insight into both normal and pathological movement control, as well as human-robot interactions. Under the assumption of stochastic optimal control, our movement predictions became more realistic when modelling neuro-musculoskeletal dynamics in more detail. Moving beyond deterministic simulations may enable coupled changes in complex, whole body movement and its neural control to be studied in both healthy and impaired movement. It is critical to consider interactions of feedforward and feedback neural control mechanisms with musculoskeletal mechanics in a highly redundant space of feasible neuromusculoskeletal solutions for movement, particularly in the presence of noise. Optimal control processes may also explain compensatory changes in neural control of movement in impaired motor control [73]. Moreover, the role of nonlinear musculoskeletal dynamics has been recently demonstrated to play a significant role in impaired motor control such as in spasticity [6]. Further, movement variability might complicate the control and design of exoskeletons and other assistive devices. Incorporating uncertainty and movement variability in simulations that are often used to generate and test ideas before implementing these in reality, could facilitate the design process of assistive devices.

## Methods

### Approximating the stochastic optimal control problem by a deterministic optimal control problem

We formulate simulations of movement in the presence of noise as stochastic optimal control problems. The system dynamics are stochastic: $\dot{x}(t) = f(x(t), e(t), w_{sys})$; with $x$ the state trajectories, $e$ the control trajectories and $w_{sys}$ a set of stochastic disturbances with zero-mean Gaussian distribution (noise) acting on the system. The control trajectories $e(t)$ are determined by the control policy:

$$e(t) = e_{ff}(t) + K(t) * y_{fb}(x(t), w_{policy}),$$

which consists of a deterministic feedforward term $e_{ff}(t)$ and a linear feedback term with the deterministic feedback matrix $K(t)$ that is multiplied with the feedback signal $y_{fb}(x(t), w_{policy})$. $w_{policy}$ is a set of stochastic disturbances, with a zero-mean Gaussian distribution, acting within the control policy. Note that when $K(t)$ is non-zero, the control trajectories $e(t)$ are stochastic, and depend on the state.

To solve the stochastic optimal control problems, we approximate the stochastic state trajectories, which are in general non-normally distributed, by normally distributed trajectories. As a result, we can describe the stochastic state trajectory by the mean state trajectory $x_{mean}(t)$ and state covariance trajectory $P(t)$. The dynamics of the mean state can be described by a deterministic approximation of the stochastic dynamics obtained by setting the disturbances to their mean value, which is zero ($w = [w_{sys}, w_{policy}] = 0$):

$$\dot{x}_{mean}(t) = f(x_{mean}(t), e(t), w = 0). \tag{1}$$

The dynamics of the state covariance can be described by the continuous Lyapunov differential equations based on a local first-order approximation of the nonlinear system dynamics around the mean state, corresponding to the propagation rules used in the extended Kalman Filter ([29,74]):

$$\dot{P}(t) = A(t)P(t) + P(t)A(t)^T + C(t)\Sigma_w' C(t)^T \tag{2}$$

$$A(t) = \left( \frac{\partial f}{\partial x}(t, x(t), e(t), w) \right)_{x(t)=x_{mean}(t)} \tag{3}$$

$$C(t) = \left( \frac{\partial f}{\partial w}(t, x(t), e(t), w) \right)_{x(t)=x_{mean}(t)} \tag{4}$$

with $\Sigma_w'$ the continuous time covariance matrix describing the noise sources. Eqs (1–4) form a deterministic approximation of the stochastic dynamics. Similarly, we can approximate the stochastic constraint functions $g(x(t))$ by a normal distribution with mean $\mu_{g(x)} = g(x_{mean})$, and standard deviation $\sigma_{g(x)} = \sqrt{\frac{\partial g}{\partial x} P(t) \frac{\partial g}{\partial x}^T}$. Using this approach, we can transform the stochastic optimal control problem into an approximate deterministic optimal control problem:

$$\min_{e_{ff}(t), K(t)} \int_{t_{start}}^{t_{final}} J_{cost}(x_{mean}(t), e_{ff}(t), K(t), P(t)) dt \text{ cost} \tag{5}$$

$$\text{subject to } \dot{x}_{mean}(t) = f(x_{mean}(t), e(t), w = 0) \text{ mean state dynamics} \tag{6}$$

$$\dot{P}(t) = A(t)P(t) + P(t)A(t)^T + C(t)\Sigma'_w C(t)^T \text{ covariance dynamics} \qquad (7)$$

$$g_i(x_{\text{mean}}(t), u(t)) + \gamma_i \sqrt{\frac{\partial g_i}{\partial x} P(t) \frac{\partial g_i^T}{\partial x}} \geq 0 \ i = 1, \dots n_g \text{ path constraints} \qquad (8)$$

with $\gamma_i$ a parameter determining the probability that the state trajectory fulfills the constraint and $n_g$ the number of path constraints. Given the Gaussian approximation of the state distribution, $\gamma_i$ would need to be infinitely large to impose the constraint over the whole distribution. In practice, we choose finite values for $\gamma_i$, where the chance of fulfilling the constraints is 95% when $\gamma_i = 2$, and 99.7% when $\gamma_i = 3$.

We solve the approximate deterministic optimal control problems using direct collocation with a trapezoidal integration scheme and mesh intervals of 10ms and solve the resulting large, but sparse nonlinear programming problems (NLP) with IPOPT [75]. We formulate all dynamics implicitly (details in S1 Text) to improve the numerical condition of the NLP [6,23,76]. We use CasADi [77] to perform automatic differentiation, which improves the accuracy of the derivative computations and might reduce the number of operations to compute gradients through use of its reverse mode [24]. For details on the numerical implementation of the optimal control problems, we refer to S1 Text. The code for simulating the presented models is publicly available at: https://github.com/tomvanwouwe1992/SOC_Paper.

## Stochastic optimal control simulations of movement

We applied our stochastic optimal control framework to two fundamental movements that have been studied extensively: standing balance and goal-directed reaching. An overview of the musculoskeletal and motor control models is provided in Figs 1 and 2. In general, we solved for both feedforward (i.e., open-loop) and feedback components (i.e. feedback gains) of the control law to perform the prescribed movement task robustly with minimal expected effort. For standing balance, we imposed task robustness by requiring the solution to be marginally stable, i.e. the state covariance was constant. For reaching, we imposed the required accuracy of the reaching movement depending on the target shape.

## Standing balance

We first give a general description of the optimal control problem, simulations and outcome measures. Dynamic equations and model parameters are described in more detail below.

### General description of standing balance simulations and outcome measures

We modeled standing in the presence of platform perturbations based on an inverted pendulum (IP) model (mass: 70kg, length: 1m) linked to a rotatable and translational platform (Fig 1). Two antagonistic Hill-type muscles with rigid tendons actuated the ankle joint, i.e., the joint connecting the pendulum to the platform. The muscle properties (maximal isometric force $F_{ISO}$, tendon slack length $l_T^s$, optimal fiber length $l_M^o$, optimal pennation angle α, damping coefficient β; described in Table 1) were taken from the soleus and tibialis anterior muscle of the OpenSim3.3 gait10dof18musc model [78]. Input to the Hill-type muscles were muscle activations ($a_{SOL}$, $a_{TA}$). Activations resulted from excitations through first-order dynamics with a time constant ($\tau$) of 150ms lumping together sensory and motor delays [79]. These muscle excitations resulted from excitations that were composed of feedforward ($e_{SOL,ff}$, $e_{TA,ff}$) and

feedback excitations ($e_{SOL,fb}$, $e_{TA,fb} = K^* y_{fb}$). The feedback excitations were a linear combination of the angle and angular velocity of the inverted pendulum with respect to the gravitational field, representing vestibular information, and of the angle and angular velocity of the inverted pendulum with respect to the platform, representing proprioceptive information. All feedback gains were constant in time. The policy noise ($w_{policy}$) consisted of Gaussian vestibular ($w_{v,q}$, $w_{v,\dot{q}}$) and proprioceptive noise ($w_{p,q}$, $w_{p,\dot{q}}$) with respective variance $\sigma^2_{v,q}$, $\sigma^2_{v,\dot{q}}$ and $\sigma^2_{p,q}$, $\sigma^2_{p,\dot{q}}$ that was added to both the vestibular and proprioceptive cues. Motor noise, which was part of the system noise ($w_{sys}$), consisted of additive motor noise ($w_{SOL}$, $w_{TA}$) with variance ($\sigma^2_{SOL}$, $\sigma^2_{TA}$) that corrupted the muscle activations. All sensory and motor noise sources were independent and values for the noise source variances are provided in Table 2.

We modeled random rotation and translation perturbations through adding additional sources of system noise ($w_{sys}$). The random platform angular position and velocity were modeled as zero mean Gaussian noise ($w_{SS,q}$, $w_{SS,\dot{q}}$) with constant variance ($\sigma^2_{SS,q}$, $\sigma^2_{SS,\dot{q}}$). We introduce the subscript 'ss' to indicate the variables related to the platform or support-surface and to avoid confusion with subscript 'p' for proprioception. To simulate random translation perturbations, we only needed to describe the platform translational acceleration as zero mean Gaussian noise ($w_{SS,trans}$) with variance ($\sigma^2_{SS,trans}$). We could ignore the random translational position and velocity of the platform as these affect neither the dynamics of the pendulum nor the proprioceptive and vestibular information (relative and absolute pendulum angles). Values for the noise source variances are provided in Table 2.

The task-goal during standing was to maintain a stable upright posture (marginal stability). Postural marginal stability, in the presence of noise, was modeled by constraining the mean angle of the pendulum to be upright with zero angular velocity, the mean state derivatives to be zero (mean pendulum position and velocity and muscle activations are constant in time), and the state covariance derivatives to be zero: $\dot{P} = 0$ (state covariance matrix was constant in time). We thus solved for a single state, rather than a state trajectory, and the corresponding control policy to maintain a stable upright posture in the presence of noise. We solved for the constant feedforward muscle excitations and feedback gains given the described constraints while minimizing expected effort. Expected effort was modeled as the expected value of the sum of muscle excitations squared:

$$E[(e_{SOL,ff} + e_{SOL,fb})^2 + (e_{TA,ff} + e_{TA,fb})^2] \tag{9}$$

which is equivalent to

$$e_{SOL,ff}{}^2 + e_{TA,ff}{}^2 + Var[e_{SOL,fb}] + Var[e_{TA,fb}] \tag{10}$$

since the deterministic feedforward terms are uncorrelated to the stochastic feedback terms

**Table 1. Muscle properties of model for perturbed standing simulations.**

| Muscle properties | | |
|---|---|---|
| | **soleus** | **tibialis anterior** |
| $F_{ISO}[N]$ | 5137 | 3000 |
| $l^s_T[m]$ | 0.2514 | 0.2228 |
| $l^o_M[m]$ | 0.0528 | 0.1028 |
| $\alpha$ [rad] | 0.4364 | 0.0873 |
| $\beta$ | 0.01 | 0.01 |

**Table 2. Noise characteristics for perturbed standing balance simulations.**

| Sensory noise | | Motor noise | | $\sigma_{SS,\dot{q}}^2\left[\left(\frac{\circ}{s}\right)^2/Hz\right]$ | $0.001^2; 0.3^2; 0.6^2; 1.2^2; 2.4^2; 4.8^2; 9.6^2$ |
|---|---|---|---|---|---|
| $\sigma_{p,q}^2[(\circ)^2/Hz]$ | $0.1^2$ | $\sigma_{SOL}^2[(-)^2./Hz]$ | $0.01^2$ | | |
| $\sigma_{p,\dot{q}}^2\left[\left(\frac{\circ}{s}\right)^2/Hz\right]$ | $0.2^2$ | $\sigma_{TA}^2[(-)^2./Hz]$ | $0.01^2$ | **Platform translations** | |
| $\sigma_{v,q}^2[(\circ)^2/Hz]$ | $0.3^2$ | **Platform rotations** | | $\sigma_{SS,trans}^2\left[\left(\frac{m}{s^2}\right)^2/Hz\right]$ | $0.001^2; 0.0175^2; 0.035^2; 0.07^2; 0.14^2; 0.28^2; 0.56^2$ |
| $\sigma_{v,\dot{q}}^2\left[\left(\frac{\circ}{s}\right)^2/Hz\right]$ | $0.6^2$ | $\sigma_{SS,q}^2[(\circ)^2/Hz]$ | $0.001^2; 0.125^2; 0.25^2; 0.5^2; 1^2; 2^2; 4^2$ | | |

and because the stochastic feedback terms ($e_{SOL,fb}$, $e_{TA,fb}$) have an expected value of zero (a derivation can be found in the S1 Text).

We performed simulations based on four different models. We performed simulations with the full feedback model, representing healthy subjects, and simulations with only proprioceptive feedback, representing vestibular loss subjects. We performed these simulations based on two muscle models, a Hill-type muscle model and a Hill-type model that was extended with short-range-stiffness (SRS). Short-range stiffness was modeled by adding a spring with activation dependent stiffness in parallel to the contractile element producing a force

$$F_{SRS} = k_{SRS} \cdot F_{iSo,max} \cdot a \cdot \left(\frac{l_m - l_{m,mean}}{l_{m,opt}}\right), \tag{11}$$

with $a$ the muscle baseline activation, $\tilde{l}_m$ the muscle fiber length, $\tilde{l}_{m,mean}$ the muscle fiber length in mean (upright) position, $\tilde{l}_{m,opt}$ the optimal muscle fiber length and $k_{SRS}$ the short-range-stiffness scaling factor, which was set to 1 [34]. We thus have four models: healthy ('healthy'), healthy with muscles including short-range-stiffness ('healthy—SRS'), vestibular loss ('VL'), vestibular loss with muscles including short-range-stiffness ('VL—SRS').

Our outcome measures were (1) body sway–described by the standard deviation of the normally distributed pendulum angle, (2) the relative contribution of proprioceptive feedback, $\frac{\|K_{proprio}\|}{\|K_{proprio}\| + \|K_{vest}\|}$, (3) the co-contraction index (CCI), computed as in [47]: $\frac{min(a_{SOL}, a_{TA})}{\max(a_{SOL}, a_{TA})} * (a_{SOL} + a_{TA})$, and (4) the contribution of expected effort from feedback to the total expected effort: $\frac{effort_{FB}}{effort_{FB} + effort_{FF}}$.

## Stochastic dynamics and model parameters

We indicated variables that are modeled as Gaussian noise in red. The state consisted of the segment angle and angular velocity, $q, \dot{q}$, the activation of soleus and tibialis anterior, $a_{SOL}$ and $a_{TA}$ and the platform angle and angular velocity, $q_{SS}$ and $\dot{q}_{SS}$:

$$\boldsymbol{x} = \begin{bmatrix} q & \dot{q} & a_{SOL} & a_{TA} & q_{SS} & \dot{q}_{SS} \end{bmatrix} \tag{12}$$

The control law was parametrized by the baseline muscle excitations, $\boldsymbol{e}_{ff}$, and the constant feedback gains, $\boldsymbol{K}$:

$$\boldsymbol{e}_{ff} = \begin{bmatrix} e_{SOL,ff} & e_{TA,ff} \end{bmatrix}; \boldsymbol{K} = \begin{bmatrix} K_{q,prop}^{SOL} & K_{\dot{q},prop}^{SOL} & K_{q,vest}^{SOL} & K_{\dot{q},vest}^{SOL} \\ K_{q,prop}^{TA} & K_{\dot{q},prop}^{TA} & K_{q,vest}^{TA} & K_{\dot{q},vest}^{TA} \end{bmatrix}, \tag{13}$$

where SOL refers to the soleus and TA refers to the tibialis anterior. The dynamics were

described by the equations of motion of the pendulum and the first order delay between excitation and activation. The equations of motion were expressed in a non-inertial reference frame by introducing fictitious forces due to the translational platform acceleration:

$$
\begin{bmatrix}
dq/dt \\
d\dot{q}/dt \\
da_{SOL,fb}/dt \\
da_{TA,fb}/dt \\
dq_{SS}/dt \\
d\dot{q}_{SS}/dt
\end{bmatrix}
=
\begin{bmatrix}
\dot{q} \\
\dfrac{mgl;sin(q)}{ml^2 + I} + \dfrac{T_{SOL+TA}}{ml^2 + I} + \dfrac{ml;sin(q)}{ml^2 + I} w_{SS,trans} \\
(e_{SOL,fb} + e_{SOL,ff} - a_{SOL})/\tau \\
(e_{TA,fb} + e_{TA,ff} - a_{TA})/\tau \\
0 \\
0
\end{bmatrix}
\tag{14}
$$

with $m$ the pendulum mass, $l$ the pendulum length, $g$ the gravity constant, $I$ the pendulum inertia, $T_{SOL+TA}$ the torque generated by the soleus and tibialis anterior. Note that the platform has a constant velocity $\frac{dq_{SS}}{dt} = 0$ and acceleration $\frac{d\dot{q}_{SS}}{dt} = 0$, meaning that the mean platform position and velocity and the platform position and velocity covariance matrix are per definition constant in time and equal to their chosen initial value. The ankle angle and angular velocity result from the difference between the pose of the segment and the platform:

$$
q_A = q - q_{SS,q}; \; \dot{q}_A = \dot{q} - \dot{q}_{SS,\dot{q}}
$$

The feedback muscle excitations ($e_{SOL,fb}$, $e_{TA,fb}$) resulted from linear feedback of the feedback signal $y_{fb}$ that consists of a proprioceptive, and vestibular ('v') signal:

$$
y_{fb} =
\begin{bmatrix}
q_A + w_{p,q} \\
\dot{q}_A + w_{p,\dot{q}} \\
q + w_{v,q} \\
\dot{q} + w_{v,\dot{q}}
\end{bmatrix}
; \;
\begin{bmatrix}
e_{SOL,fb} \\
e_{TA,fb}
\end{bmatrix}
= K; y_{fb}
\tag{15}
$$

Note that we assumed here that the platform's mean linear and angular position, velocity and acceleration were constant in time with constant variance. This differs from driving the platform acceleration with a zero-mean Gaussian input signal, which would lead to a monotonic increase of the velocity and position variance in time. Instead, we assumed that a more clever platform controller was used.

The torque generated by the soleus and tibialis anterior muscles is a function of their forces and moment arms:

$$
T_{SOL+TA} = F_{SOL} \cdot d_{SOL}(q_A) + F_{TA} \cdot d_{TA}(q_A)
\tag{16}
$$

with $d_{SOL}(q_A)$, $d_{TA}(q_A)$ the soleus and tibialis anterior moment arms, which depend on the ankle angle $q_A$. Muscle forces depend on muscle activation and through the force-length-velocity properties of the muscle also on the muscle length ($l_{SOL}$, $l_{TA}$) and velocity ($\dot{l}_{SOL}$, $\dot{l}_{TA}$), which in turn are a function of the ankle angular position and velocity $q_A, \dot{q}_A$ :

$$
\begin{bmatrix}
F_{SOL} \\
F_{TA}
\end{bmatrix}
=
\begin{bmatrix}
F_{ISO,SOL} \cdot [(a_{SOL,fb} + a_{SOL,base} + w_{SOL}) \cdot f_l(l_{SOL}) \cdot f_v(l_{SOL}, \dot{l}_{SOL}) + f_{p,SOL}(l_{SOL})] \\
F_{ISO,TA} \cdot [(a_{TA,fb} + a_{TA,base} + w_{TA}) \cdot f_l(l_{TA}) \cdot f_v(l_{TA}, \dot{l}_{TA}) + f_{p,TA}(l_{TA})]
\end{bmatrix}
\tag{17}
$$

with $f_l$ the active muscle force-length relationship, $f_v$ the muscle force-velocity relationship,

and $f_p$ the passive muscle force-length relationship. The active force-length, force-velocity, and passive force-length relationships are described in [22].

Muscle-tendon lengths were approximated by the sum of a linear function, a sine, and a constant offset ($l_{MT} = a\cdot q + b\cdot \sin(c\cdot q) + d$) with a, b, c and d estimated by minimizing the least square error between this approximation and the muscle lengths obtained from the OpenSim gait10dof18musc model [75]. The moment-arms are computed as the derivatives of the muscle-tendon lengths with respect to the angle: $d_{muscle} = a + b\cdot c\cdot\cos(c\cdot q)$ [78].

Noise characteristics (summarized in Table 2) were based on preliminary simulations and experimental data [36]. Motor noise, added to muscle activations, had a standard deviation of 1% of the maximal signal based on force fluctuation measurements in different isometric tasks [80,81], where a coefficient of variation between 1–5% was found. These measurements quantify motor noise indirectly as force-tracking errors might have other origins as well and we therefore selected the lower end of the measured variability to model motor noise. The relative values of proprioceptive and vestibular noise were selected such that the relative contribution of proprioceptive feedback, $\frac{\|K_{prop}\|}{\|K_{prop}\| + \|K_{vest}\|}$, was between 0.7–0.8 during optimal unperturbed standing in agreement with values identified from experiments by Peterka [32]. The absolute values for sensory noise were selected such that during optimal unperturbed standing body sway, defined as the standard deviation of the pendulum angle, was ~0.3˚, a typical value found in experiments of quiet standing in healthy subjects [78]. The values that determine the platform rotations, $\sigma^2_{SS,q}$, $\sigma^2_{SS,\dot{q}}$, were selected to mimic the rotational perturbations applied in [36]. In these experiments, the platform angular position, velocity and acceleration were not Gaussian. Here, we approximated the non-Gaussian experimental platform movements by zero-mean Gaussian platform movements with a standard deviation of half the amplitude of the experimental perturbations. The variance of the translational accelerations $\sigma^2_{SS,trans}$ were determined such that the healthy model for the maximal accelerations, under optimal control, reached a standard deviation of the ankle angle of 4˚, a value that is typically not exceeded in continuous translation perturbation experiments [82,83].

The time unit appears in the noise variance ([/Hz]) or ([.s]) to describe the power spectral density of a continuous-time Gaussian noise. If we perform a numerical integration and thus move to a discrete-time description of continuous noise the unit of time disappears by dividing the variance by the integration interval length (expressed in seconds). This makes sense if we reflect about a forward integration of a 1D point mass where the velocity has a continuous variance of e.g. 2 (m/s)$^2$/Hz. If we perform the numerical integration over an interval of 1s the discrete variance ($\Sigma$) is 2 (m/s)$^2$, we find that the variance of the position after 1s is equal to 2m$^2$: $P_{k+1} = P_k + dt * \Sigma * dt'$ ➜ $P_{k+1} = 0 + 1[s] * 2[(m/s)2] * 1[s] = 2m^2$. If we perform the numerical integration over 1s using a time step of 0.1s the discrete variance is 20 (m/s)$^2$. We obtain $P_{0.1s} = 0 + 0.1[s] * 20[(m/s)^2] * 0.1[s] = 0.2m^2$. We obtain for $P_{0.2s} = 0.2[m^2] + 0.1[s] * 20[(m/s)^2] * 0.1[s] = 0.4m^2$. Finally, $P_{1s} = 2m^2$.

## Goal-directed reaching

**General description of reach simulations and outcome measures.** We modeled four reaching tasks based on a two-segment model, where the segments represent the upper and lower arm and are connected by hinge joints (Fig 2). The arm model was driven by six Hill-type muscles with rigid tendons consisting of a uni-articular agonist-antagonist pair for the shoulder (anterior and posterior deltoid) and the elbow (brachialis and lateral triceps) and a bi-articular agonist-antagonist pair spanning both the elbow and the shoulder (biceps and long triceps).The muscle properties (maximal isometric force $F_{ISO}$, normalized maximal contraction velocity $v_{max}$, and damping coefficient β; described in Table 3) were identical to the

model used and reported in [20] and are reported in Table 3. The muscles were stimulated by muscle excitations ($e_{BRACH}$, $e_{LATTRI}$, $e_{ANTDEL}$, $e_{POSTDEL}$, $e_{BIC}$, $e_{LONGTRI}$) that were a sum of feedforward ($e_{ff}$) and feedback excitations ($e_{fb} = K^* y_{fb}$). Feedback excitations consisted of linear time-varying feedback of the end effector (hand) position and velocity with respect to the nominal end effector kinematics. The nominal end effector kinematics was the end effector kinematics due to the feedforward excitations in the absence of sensory and motor noise. Activations were related to excitations through first-order dynamics with a time constant ($\tau$) of 150ms lumping together sensorimotor delays. Policy or sensory noise was modeled by additive Gaussian noise on the end effector position ($w_{EE_x}$, $w_{EE_y}$) and velocity ($w_{\dot{E}E_x}$, $w_{\dot{E}E_y}$) with respective variance ($\sigma^2_{EE,x}$, $\sigma^2_{EE,y}$, $\sigma^2_{EE,\dot{x}}$, $\sigma^2_{EE,\dot{y}}$). The noisy end effector positions and velocities were input to the feedback law. System noise consisted of motor noise that was modeled by Gaussian additive noise added to each joint torque ($w_s$, $w_e$) with variance ($\sigma^2_s$, $\sigma^2_e$).

The task-goal was to perform a point-to-point reaching movement over a distance of 25cm in 0.8s with a pre-defined chance of the end effector ending up within the target. Reaching accuracy was imposed by limiting the variance of the horizontal and vertical end effector position depending on the task requirements.

1. Reaching towards a small circular target (circle) was modeled by constraining the standard deviation of the end effector end-point horizontal and vertical positions to be smaller than 0.4cm.

2. Reaching towards a horizontal bar (bar) was modeled by constraining the standard deviation of the end effector end-point vertical position to be smaller than 0.4cm

3. Reaching towards a circular target in the presence of an obstacle was modeled by imposing the standard deviation of the end effector end-point horizontal and vertical positions to be smaller than 0.4cm and the standard deviation of end effector horizontal position during the second part (>0.25s) of the reaching trajectory to be smaller than 0.4cm

4. Reaching towards a circular target in the presence of a divergent force field of 200N/m was modeled by constraining the standard deviation of the end effector end-point horizontal and vertical positions to be smaller than 0.4cm.

We solved for optimal control policies that minimized expected effort,

$$E\left[\int \left[e_{BRACH}(t)^2 + e_{LATTRI}(t)^2 + e_{ANTDEL}(t)^2 + e_{POSTDEL}(t)^2 + e_{BIC}(t)^2 + e_{LONGTRI}(t)^2\right]dt\right], \quad (18)$$

while fulfilling the task requirements for each of the specific tasks. We then used these optimal control policies to perform 100 forward simulations of unperturbed and perturbed reaching. The simulated extension perturbations matched the perturbations in the experiments described by Nashed et al. [8] and are shown in Fig 2C.

**Table 3. Musculoskeletal properties of model for goal-directed reaching.**

| | Skeletal properties | | | Muscle properties | | | | | |
|---|---|---|---|---|---|---|---|---|---|
| | Upper arm | Forearm | | BRACH | LATTRI | ANTDEL | POSTDEL | BIC | LONGTRI |
| $m$ [kg] | 1.4 | 1.0 | $F_{ISO}$[N] | 572 | 445 | 700 | 382 | 159 | 318 |
| $l$ [m] | 0.3 | 0.33 | $\bar{v}_{M,max}$ | 10 | 10 | 10 | 10 | 10 | 10 |
| $I$[kg.m$^2$] | 0.025 | 0.045 | β | 0.01 | 0.01 | 0.01 | 0.01 | 0.01 | 0.01 |

We computed the accuracy for the different optimal control policies in perturbed and unperturbed reaching by computing the 95% confidence ellipses of the end-point positions of the end effector for the 100 simulations. We computed the corrective muscle activations for perturbed reaching by subtracting the mean muscle activations during unperturbed reaching from the muscle activations for each of the 100 perturbed reaching simulations. We computed the mean and standard deviation of these corrective muscle activations to analyze corrective behavior at the muscle level. We computed the co-contraction index throughout the reaching movement for each of the joints by averaging the joint-specific CCI over time. We computed the CCI as in [47]: $\frac{min(a_{flex}(t),a_{ext}(t))}{\max(a_{flex}(t),a_{ext}(t))} * (a_{flex}(t) + a_{ext}(t))$ where $a_{flex}$ and $a_{ext}$ are the activations of the flexor and extensor muscles of the antagonistic pairs.

## Stochastic dynamics and model parameters

We indicated variables that are described as constant zero-mean Gaussian noise in red. The state consisted of the joint angular positions $(q_s, q_e)$ and velocities $(\dot{q}_s, \dot{q}_e)$, and the activations of the anterior deltoid, the posterior deltoid, the biceps and the triceps muscles ($a_{BRACH}$, $a_{LAT\text{-}TRI}$, $a_{ANTDEL}$, $a_{POSTDEL}$, $a_{BIC}$, $a_{LONGTRI}$).

$$x = \begin{bmatrix} q_s & q_e & \dot{q}_s & \dot{q}_e & a_{BRACH} & a_{LATTRI} & a_{ANTDEL} & a_{POSTDEL} & a_{BIC} & a_{LATTRI} \end{bmatrix} \quad (19)$$

The control law was parametrized by the feedforward muscle excitation trajectories $\boldsymbol{e}_{ff}(t)$ and the time-varying feedback gains, $\boldsymbol{K}(t)$:

$$\boldsymbol{e}_{ff}(t) = \begin{bmatrix} e_{ff,BRACH}(t) \\ e_{ff,LATTRI}(t) \\ e_{ff,ANTDEL}(t) \\ e_{ff,POSTDEL}(t) \\ e_{ff,BIC}(t) \\ e_{ff,LONGTRI}(t) \end{bmatrix} ; \boldsymbol{K}(t) = \begin{bmatrix} K_{q_s}^{BRACH}(t) & K_{q_e}^{BRACH}(t) & K_{\dot{q}_s}^{BRACH}(t) & K_{\dot{q}_e}^{BRACH}(t) \\ K_{q_s}^{LATTRI}(t) & K_{q_e}^{LATTRI}(t) & K_{\dot{q}_s}^{LATTRI}(t) & K_{\dot{q}_e}^{LATTRI}(t) \\ K_{q_s}^{ANTDEL}(t) & K_{q_e}^{ANTDEL}(t) & K_{\dot{q}_s}^{ANTDEL}(t) & K_{\dot{q}_e}^{ANTDEL}(t) \\ K_{q_s}^{POSTDEL}(t) & K_{q_e}^{POSTDEL}(t) & K_{\dot{q}_s}^{POSTDEL}(t) & K_{\dot{q}_e}^{POSTDEL}(t) \\ K_{q_s}^{BIC}(t) & K_{q_e}^{BIC}(t) & K_{\dot{q}_s}^{BIC}(t) & K_{\dot{q}_e}^{BIC}(t) \\ K_{q_s}^{LONGTRI}(t) & K_{q_e}^{LONGTRI}(t) & K_{\dot{q}_s}^{LONGTRI}(t) & K_{\dot{q}_e}^{LONGTRI}(t) \end{bmatrix} \quad (20)$$

The dynamics were described by the equations of motion and the first order delay between excitations and activations:

$$\begin{bmatrix} \frac{d\boldsymbol{q}}{dt} \\ \frac{d\dot{\boldsymbol{q}}}{dt} \\ \frac{d\boldsymbol{a}}{dt} \end{bmatrix} = \begin{bmatrix} \dot{\boldsymbol{q}} \\ M(\boldsymbol{q})^{-1}(C(\boldsymbol{q},\dot{\boldsymbol{q}}) + \boldsymbol{T}_M + \begin{bmatrix} W_s \\ w_e \end{bmatrix}) \\ (\boldsymbol{e} - \boldsymbol{a})/\tau \end{bmatrix} \quad (21)$$

with $M(\boldsymbol{q})$ the mass-matrix of the arm model, $C(\boldsymbol{q},\dot{\boldsymbol{q}})$ the term describing the Coriolis forces, $\boldsymbol{T}_M$ the shoulder and elbow joint torques generated by the muscles and $w_s$, $w_e$ the stochastic torque acting at shoulder and elbow. The total muscle excitations were the result of feedforward and feedback control, where the feedback signal is the noisy $(w_{EE_x}, w_{EE_y}, w_{\dot{EE}_x}, w_{\dot{EE}_y})$ deviation from the end effector position and velocity with respect to the expected or mean end

effector positions and velocity:

$$
\boldsymbol{y}_{fb} = \begin{bmatrix} EE_x + w_{EE_x} \\ EE_y + w_{EE_y} + w_{EE_x} EE_y + w_{EE_y} \\ EE_x \end{bmatrix} - \boldsymbol{EE}_{ref}; \; \boldsymbol{e} = \boldsymbol{e}_{ff} + \boldsymbol{K} \cdot \boldsymbol{y}_{fb}. \tag{22}
$$

The end effector positions ($EE_x$, $EE_y$) and velocities ($\dot{EE}_x$, $\dot{EE}_y$) can be computed from the joint positions and velocities:

$$
\begin{bmatrix} EE_x \\ EE_y \, EE_y \\ EE_x \end{bmatrix} = f_{kin}(\boldsymbol{q}, \dot{\boldsymbol{q}}) \tag{23}
$$

The reference end effector trajectory ($\boldsymbol{EE}_{ref}$) is the end effector trajectory in the absence of noise, which can thus be computed from the mean joint trajectories.

$$
\boldsymbol{EE}_{ref} = f_{kin}(\boldsymbol{q}_{mean}, \dot{\boldsymbol{q}}_{mean})
$$

The shoulder and elbow torques generated by the different muscles depends on muscle forces and moment arms, which in turn depend on the shoulder and elbow joint angles:

$$
T_M = \begin{bmatrix} F_{ANTDEL}; d_{ANTDEL}(q_s) + F_{POSTDEL} \cdot d_{POSTDEL}(q_s) + F_{BIC} \cdot d_{BIC,s}(q_s) + F_{LONGTRI} \cdot d_{LONGTRI,s}(q_s) \\ F_{BIRACH}; d_{BRACH}(q_e) + F_{LATTRI} \cdot d_{LATTRI}(q_e) + F_{BIC} \cdot d_{BIC,e}(q_e) + F_{LONGTRI} \cdot d_{LONGTRI,e}(q_e) \end{bmatrix}, \tag{24}
$$

with
$d_{ANTDEL}(q_s), d_{POSTDEL}(q_s), d_{BRACH}(q_e), d_{LATTRI}(q_e), d_{LONGTRI,s}(q_s), d_{LONGTRI,e}(q_e), d_{BIC,e}(q_e), d_{BIC,s}(q_s)$
the muscle moment arms depending on the articulated joint angles. Muscle forces depend on muscle activation and through the force-length-velocity properties of the muscle also on the normalized muscle fiber length ($l_{BRACH}$, $l_{LATTRI}$, $l_{ANTDEL}$, $l_{POSTDEL}$, $l_{BIC}$, $l_{LONGTRI}$) and velocity ($\dot{l}_{BRACH}$, $\dot{l}_{LATTRI}$, $\dot{l}_{ANTDEL}$, $\dot{l}_{POSTDEL}$, $\dot{l}_{BIC}$, $\dot{l}_{LONGTRI}$), which in turn were a function of skeleton positions and velocities ($q_s, q_e, \dot{q}_s, \dot{q}_e$):

$$
\begin{bmatrix} F_{BRACH} \\ F_{LATTRI} \\ F_{ANTDEL} \\ F_{POSTDEL} \\ F_{BIC} \\ F_{LONGTRI} \end{bmatrix} = \begin{bmatrix} F_{ISO,BRACH} \cdot [a_{BRACH} \cdot f_l(l_{BRACH}) \cdot f_v(l_{BRACH}, \dot{l}_{BRACH}) + f_p(l_{BRACH})] \\ F_{ISO,LATTRI} \cdot [a_{LATTRI} \cdot f_l(l_{LATTRI}) \cdot f_v(l_{LATTRI}, \dot{l}_{LATTRI}) + f_p(l_{LATTRI})] \\ F_{ISO,ANTDEL} \cdot [a_{ANTDEL} \cdot f_l(l_{ANTDEL}) \cdot f_v(l_{ANTDEL}, \dot{l}_{ANTDEL}) + f_p(l_{ANTDEL})] \\ F_{ISO,POSTDEL} \cdot [a_{POSTDEL} \cdot f_l(l_{POSTDEL}) \cdot f_v(l_{POSTDEL}, \dot{l}_{POSTDEL}) + f_p(l_{POSTDEL})] \\ F_{ISO,BIC} \cdot [a_{BIC} \cdot f_l(l_{BIC}) \cdot f_v(l_{BIC}, \dot{l}_{BIC}) + f_p(l_{BIC})] \\ F_{ISO,LONGTRI} \cdot [a_{LONGTRI} \cdot f_l(l_{LONGTRI}) \cdot f_v(l_{LONGTRI}, \dot{l}_{LONGTRI}) + f_p(l_{LONGTRI})] \end{bmatrix} \tag{25}
$$

with $f_l$ the active muscle force-length relationship, $f_v$ the muscle force-velocity relationship, and $f_p$ the passive muscle force-length relationship. The active force-length, force-velocity, and passive force-length relationships are described in [22].

Normalized muscle fiber lengths were approximated by the sum of linear functions and sines of the joint angles added to a constant offset:
($l_M = a_s \cdot q_s + \frac{b_s}{c_s} \cdot \sin(c_s \cdot q_s) + a_e \cdot q_e + \frac{b_e}{c_e} \cdot \sin(c_e \cdot q_e) + d$). The muscle-moment arms were computed as scaled versions of the derivatives of the normalized fiber lengths with respect to the joint angle: $d_s = s_s(a_s + b_s \cdot \sin(c_s \cdot q_s))$; $d_e = s_e(a_e + b_e \cdot \sin(c_e \cdot q_e))$, where the scaling

**Table 4. Noise characteristics for goal-directed reaching simulations.**

| Sensory noise | | Motor noise | |
|---|---|---|---|
| $\sigma^2_{EE,x}[(mm)^2/Hz]$ | $0.6^2$ | $\sigma^2_{SH}[(Nm)^2/Hz]$ | $0.05^2$ |
| $\sigma^2_{EE,y}[(mm)^2/Hz]$ | $0.6^2$ | $\sigma^2_{EL}[(Nm)^2/Hz]$ | $0.05^2$ |
| $\sigma^2_{EE,\dot{x}}\left[\left(\frac{mm}{s}\right)^2./Hz\right]$ | $4.8^2$ | | |
| $\sigma^2_{EE,\dot{y}}\left[\left(\frac{mm}{s}\right)^2./Hz\right]$ | $4.8^2$ | | |

factors serve to denormalize the muscle-moment arms. The coefficients parametrizing the muscle-tendon lengths and moment arms ($a_s$, $b_s$, $c_s$, $s_s$, $a_e$, $b_e$, $c_e$, $s_e$, and $d$) were estimated to match the data provided in [20].

We computed the variance of the end effector position and velocity in the horizontal and vertical directions, used to define accuracy constraints, based on the following equations:

$$Var(EE_x(t), EE_y(t), \dot{EE}_x(t), \dot{EE}_y(t)) = trace$$

$$\left( \left( \frac{\partial f_{kin}(\boldsymbol{q}(t), \boldsymbol{q}(t))}{\partial \boldsymbol{x}} \right)_{\boldsymbol{q}_{mean}(t), \dot{\boldsymbol{q}}_{mean}(t)} \boldsymbol{P}(t) \left( \frac{\partial f_{kin}(\boldsymbol{q}(t), \dot{\boldsymbol{q}}(t))}{\partial \boldsymbol{x}} \right)_{\boldsymbol{q}_{mean}(t), \dot{\boldsymbol{q}}_{mean}(t)}' \right) \quad (26)$$

with $\boldsymbol{P}(t)$ the covariance matrix used to approximate the stochastic state.

Noise characteristics summarized in Table 4 are based on preliminary simulations and experimental data. Motor noise was defined as a noisy torque actuator with zero mean and a variance power spectral density of 0.05(Nm)$^2$/Hz. The absolute values for sensory noise were selected such that during optimal unperturbed reaching to the circle target an end-point accuracy of 0.4cm was achievable but an accuracy of 0.2cm was not [8].

## Supporting information

**S1 Text. Describes additional results, motivation and consequences on assuming rigid tendons and more details on the implemented method.**
(PDF)

## Author Contributions

**Conceptualization:** Tom Van Wouwe, Lena H. Ting, Friedl De Groote.

**Data curation:** Tom Van Wouwe.

**Formal analysis:** Tom Van Wouwe.

**Funding acquisition:** Tom Van Wouwe.

**Investigation:** Tom Van Wouwe, Friedl De Groote.

**Methodology:** Tom Van Wouwe, Friedl De Groote.

**Project administration:** Tom Van Wouwe, Friedl De Groote.

**Resources:** Friedl De Groote.

**Software:** Tom Van Wouwe.

**Supervision:** Lena H. Ting, Friedl De Groote.

**Validation:** Tom Van Wouwe.

**Visualization:** Tom Van Wouwe.

**Writing – original draft:** Tom Van Wouwe, Friedl De Groote.

**Writing – review & editing:** Tom Van Wouwe, Lena H. Ting, Friedl De Groote.

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
