## [Decision Letter · Decision Letter 0]

3 Sep 2021

Dear Mr Van Wouwe,

Thank you very much for submitting your manuscript "An approximate stochastic optimal control framework to simulate nonlinear neuromusculoskeletal models in the presence of noise" for consideration at PLOS Computational Biology.

As with all papers reviewed by the journal, your manuscript was reviewed by members of the editorial board and by several independent reviewers. The reviewers were broadly enthusiastic about the approach and its potential to extend the scope of problems for which optimal control policies can be determined and compared with data. Two reviewers raised some important concerns, however, related to the presentation of the work, technical details of the models, and the relationship of the proposed approach to other recently published methods (e.g. Berret and Jean, PLOS Computational Biology, 2020). In light of the reviews (below this email), we would like to invite the resubmission of a significantly-revised version that takes into account the reviewers' comments.

We cannot make any decision about publication until we have seen the revised manuscript and your response to the reviewers' comments. Your revised manuscript is also likely to be sent to reviewers for further evaluation.

Sincerely,

Adrian M Haith

Associate Editor

PLOS Computational Biology

Wolfgang Einhäuser

Deputy Editor

PLOS Computational Biology

Reviewer's Responses to Questions

**Comments to the Authors:**

Reviewer #1: This article presents an approximate stochastic optimal control framework for investigating feedforward and feedback control in nonlinear neuromusculoskeletal systems with sensorimotor noise. Several simulations are performed for standing and reaching tasks with perturbations and the results are compared to previous experimental data.

Overall, the paper is well-written and develops interesting ideas about the modeling of sensorimotor control. It makes original contributions by applying the framework to standing and using state-of-the-art neuromusculoskeletal models for instance. However, several points need to be addressed to complete the literature, improve the consistency of the modeling work, and clarify its implications for the neural control of movement.

Main comments

First, it must be noted that recent papers developed and applied a very similar approximate stochastic optimal control framework to arm reaching with noisy nonlinear musculoskeletal models (see references 1-3 below). Currently, these papers are not referenced. Due to the similarity with these approaches, the authors should clearly stress and discuss the novelty of their results with respect to these studies as well as revise several statements throughout the paper accordingly.

Second, there are some inconsistencies in the modeling work. The general frameworks of the main text and of the SI text do not correspond exactly, and some discrepancy between the general framework and the specific examples can be noted. For instance, in the standing task, the feedforward control acts at the level of muscle activations (a_base) but both the feedforward and feedback control terms should act at the level of muscle excitations if we refer to the general framework (as done for reaching). Similarly, motor noise should be introduced at the level of muscle excitations if we refer to the general framework described in SI, but both in the standing and reaching tasks it is introduced at the level of muscle force/torque (yet it fits with the framework described in the main text, L. 167). Furthermore, the proposed framework seems to use a reference trajectory (pref). However, pref(t) is not mentioned in the beginning of the Results but appears later in the Methods and in the SI text. These inconsistencies should be fixed to have a general framework that matches the specific examples. This also raises specific questions.

In the standing task, can the authors use some e_ff(t) (feedforward muscle excitations) rather than a_base (feedforward muscle activations) as control? Does it influence the findings about co-contraction? Also, why assuming constant controls in the standing task whereas the framework allows for time-varying controls (L. 621-622)? Could it be possible to introduce motor noise on muscle excitations directly (as suggested in SI) rather than on muscle forces/torques and, again, would it influence the results? Finally, as pref(t) seems to be a deterministic function of the mean trajectory in simulations (L. 750-751), does pref(t) need to be explicitly part of the optimization or could yfb be defined as yfb=h(x,xmean)? If the framework really needs pref(t), its nature and role must be better explained from the beginning.

Third, the way feedback is modeled in the present paper may be somewhat confusing. This could explain the failure to predict a significant muscle co-contraction in the simulation of the divergent force field experiment (discussed L. 467-489). The reason for this failure thus deserves to be clarified because previous simulations assuming feedforward-only control had predicted muscle co-contraction for unstable tasks with a similar framework (1). Here the current lack of co-contraction might be related to an overestimated efficiency of high-level feedback control in this task because state estimation processes and sensory delays are not explicitly modeled. Although the authors explain that the time constant (tau) of the first-order muscle dynamics “lumps together sensory and motor delays”, it is still different from a feedback controller using delayed sensory measurements. Here, yfb(t) is known instantaneously from the current state x(t) and it thus affects muscle excitations (control) instantaneously. The constant tau mainly specifies that muscle excitations cannot modify muscle force instantaneously but does not specify that the controller has no access to the current state (and must infer it). Thus, in a divergent force field, the brain may not know the current hand’s state as accurately as assumed by the present modeling, which could explain why muscle co-contraction is observed in experimental data. Therefore, would the model predict more muscle co-contraction if one assumes more uncertainty about yfb(t) in the divergent force field task or considers explicitly a delayed feedback signal yfb(t-delay)?

By extension, it is unclear whether the modeled feedback indirectly captures the effects of long-latency or short-latency reflexes, or a mix of both. The fact that the feedback signal yfb involves task-level variables (e.g., end-effector position/velocity) points toward a supraspinal processing (thus long-latency reflexes). However, would the results change if feedback signals involved only joint positions/velocities (i.e., q and qdot directly, without the forward kinematics mapping)? If they don’t, together with the absence of explicit modeling of sensory delays and state estimation process, this could rather point toward short-latency spinal reflexes, the gains of which increase with muscle co-contraction as noted by the authors (L. 487-489). If these considerations can affect the interpretation of K(t) and its contribution to impedance control in the proposed model, this should be better tested and stressed throughout the paper.

Other comments

Hand velocity profiles should be depicted to fully visualize trajectories in the reaching tasks. Also, it could be interesting to plot the optimal deterministic controls (in a supplementary file if needed).

Could you please detail the derivation of Eq. 28 from Eq. 1 in the SI? The reason why cross terms disappear was not clear to me. Also, is Eq. 38 in SI based on a Taylor expansion?

Note that the SI file still contains some comments from the authors. I suggest to carefully review all equations within. I did not check all equations there because they are more secondary to the article, but I am not sure that M_k is defined in Eq. 30 and the following.

Throughout the paper, the modeling work involves stochastic differential equations but uses the formalism of ordinary differential equations. Why?

Figures 2 and 3 seem to have been inverted. Please check the captions as well. In Fig 2B, please indicate the experimental conditions for each graph.

L. 527: the dynamics was denoted f(x,e,w) in the Results and becomes f(x,u,w) here. Why not introducing u(t) already from the Results?

L. 537: Why is there a prime in Eq. 2 for sigma_w? If it is the transposition operator, please note that it is also denoted by T elsewhere.

In Eq. 5 (L. 541), the expectation is not needed if all the quantities are deterministic. Please note that J is already used L. 166 to denote the integral cost (not integrand).

Please indicate the time interval in the integrals (e.g. L. 166), and mention if the optimal control problems are considered in finite horizon or if the framework is also considered in infinite horizon (as asymptotic stability is mentioned in the standing task?).

L. 403-405: this statement is not true if we consider the references below (1-3)

L. 490-492: but see previous works (1-3)

L. 501-504: this was already observed in (1)

L. 592-596: I did not understand this paragraph well. Is it to specify the constraints of the optimal control problem? Why using these assumptions that seem to be quite strong? Please clarify.

L. 689: why not considering biarticular muscles since the model and methods allowed it? Then, the authors could have directly reused existing neuromechanical models without having to adjust muscle parameters. I did not get the reason why such a simplification was implemented here.

L. 745: "Tmotor the stochastic torque acting at shoulder and elbow resulting from motor noise ws,we.". This sentence sounds strange as motor noise is added to Tmotor. Please verify.

References

1. https://doi.org/10.1371/journal.pcbi.1007414

2. https://doi.org/10.1371/journal.pcbi.1009047

3. https://doi.org/10.1016/j.automatica.2020.108874

Reviewer #2: 1. Lines 45, 46. Suggest ‘. . . to be identified since task constraints . . . penalty terms in a cost function.’

2. Line 64, Suggest ‘. . . to aid . . .’

3. Lines 105. 106. There are many mentions of ‘collocation’ and ‘shooting methods’ throughout the manuscript beginning on these lines. While the given reference [5] provides a window into learning more about what exactly are these methods, this paper would be more understandable if some definition or description of what is meant by these terms are provided. Perhaps an appropriate place would be in the Supplementary material or possibly the Discussion.

4. Line 127. References [27-[29] are given for Houska. Is this correct? Or should it be [25]-[27]?

5. Line 194. This is the first mention of short-range stiffness. The acronym SRS should be defined here.

6. Line 200. Should be ‘. . . noise was added to . . .’

7. Line 222. Reference [47] is a reference to a very general neuroscience textbook. Please use a more appropriate reference like [31]

8. Line 229. CCI was not previously defined and is only defined in the Figure 1 legend. It should be defined here. Similarly, SRS was not defined previously. Also, the labels ‘healthy-SRS’ and ‘VL-SRS’ used here and in Figure 1 is somewhat confusing since the ‘-‘ could be interpreted as a minus sign in the sense that the model is subtracting out the SRS component rather than including it. Suggest maybe ‘healthy with SRS’ and ‘VL with SRS’ be used here and in Figure 1. Also, this labeling is used in the Methods section lines 609, 610.

9. Lines 243, 244. This is an interesting prediction regarding a shift toward increased use of vestibular information for balance with increasing platform translation amplitude. This is an example of model prediction that could potentially be tested in future experiments to test the validity of the prediction (and the methods used to make the prediction) as well as revealing something new about balance control. You might want to highlight this in the Discussion.

10. Figure 1. For the experimental rotation results showing in part A, there are experiment sensory weight measures (the Kproprio/( Kproprio+Kvest) measure) from four VL subjects as a function of rotation magnitude (Figure 10 in reference [31]). These could be shown and would be in agreement with the model-predicted results.

11. Figures 2 and 3. I note that in the reviewer download of the manuscript pdf Figures 2 and 3 were reversed, but not in the pre-print pdf.

12. Figure 2. In part B the ‘Reaching accuracy’ plots should be centered below their respective reaching trajectories.

13. Figure 3C. The vertical line indicating the perturbation time for the Posterior Deltoid is missing.

14. Line 385. Should be ‘. . . to trade them off . . .’

15. Equation 8. In comparison to equation 10 in the Supplementary Information, it’s not clear why the lambda variable in equation 8 has an ‘i’ subscript. The information given at the end of the following paragraph suggest that lambda would have a fixed value.

16. Line 648. Should be ‘Table 2’.

17. Line 656. No et al. in reference [31].

18. Line 671 Table description. It’s unusual to have such a long table description. Suggest that this description would be better made a part of the text since it also applies to the reaching simulations (or any system that is simulated).

19. Line 580 and 696. Is this ‘first-order dynamics’ with 150ms time constant meant to represent or approximate actual time delays in a closed loop feedback system because it’s not possible to represent a true time delay in their model formulation? If so, it would be informative for the authors to state whether their methods allow for the inclusion of a true time delay and thus necessitating use of some approximation.

20. Lines 731, 732. This is text left over from the rotation section and should be deleted.

21. Reference section. There are numerous references with incomplete information including 2, 3, 18, 25, 27, 28, 54, 56, 63, and 65.

Supplementary Information comments:

1. First line after equation 44. Should be ‘. . . consider the first term in . . .’

2. Section (2) below equation 45. Suggest ‘One could argue that delta must be chosen to be a smaller value, but . . .’

3. Incomplete supplement references include 1, 5, and 9.

Reviewer #3: See attachment.

**Have the authors made all data and (if applicable) computational code underlying the findings in their manuscript fully available?**

Reviewer #1: Yes

Reviewer #2: **No: **The experimental data discussed in the manuscript are only examples from previous studies by others and not original data. So the presented experimental data is not an issue. The manuscript provides detailed descriptions of the methods and equations that comprise their methods but there is no explicit computer code that implements any of their examples. It would be hard, but not impossible, for another researcher to start from scratch to implement their method so actual example code would be beneficial.

Reviewer #3: **No: **The manuscript includes a statement that code will be made available via Github. No link was provided yet.

PLOS authors have the option to publish the peer review history of their article (what does this mean?). If published, this will include your full peer review and any attached files.

Reviewer #1: **Yes: **Bastien Berret

Reviewer #2: No

Reviewer #3: **Yes: **Antonie J. van den Bogert
---

## [Decision Letter · Decision Letter 1]

16 Mar 2022

Dear Mr Van Wouwe,

Thank you very much for submitting your manuscript "An approximate stochastic optimal control framework to simulate nonlinear neuromusculoskeletal models in the presence of noise" for consideration at PLOS Computational Biology. As with all papers reviewed by the journal, your manuscript was reviewed by members of the editorial board and by several independent reviewers. The reviewers appreciated the attention to an important topic. Based on the reviews, we are likely to accept this manuscript for publication. However, the reviewers pointed out a number of minor issues that should be addressed first.

Sincerely,

Adrian M Haith

Associate Editor

PLOS Computational Biology

Wolfgang Einhäuser

Deputy Editor

PLOS Computational Biology

[LINK]

Reviewer's Responses to Questions

**Comments to the Authors:**

Reviewer #1: The authors have provided satisfactory answers to most of my concerns. The article now provides a clearer contribution to the modeling of motor control, which will hopefully trigger future developments in this area.

Please note that the paper still contains a number of typos that should be corrected before publication.

For instance, in the main text:

Fitt’s law -> Fitts’ law

‘to perform a point-to-point reaching movement over a distance of 25cm in 0.5s’ -> apparently it was rather 0.8s.

Corriolis -> Coriolis

Please check Eq. 25 (Fiso and activation seem to be inverted)

Eq. 2627 is misnumbered

And in the Supplementary Information:

missing transpose in Eq. 7 and below (for the term in C(t))

please correct the ‘dynamics wherethe controls:: ‘

‘reference trajectory ()’ is still mentioned

Reviewer #2: This reviewer has only a few minor comments. The manuscript pdf did not include line numbers or page numbers. My references to page numbers below begin with the title page being page 1.

Minor comments:

1) Page 4. The references [35] [36] apparently should be [34] [35].

2) Page 4, three lines below the above references, suggest eliminating the first ‘the’ in the sentence so the sentence reads ‘This does not seem to yield an optimal solution as during the optimization . . .’

3) The SRS acronym is still not clearly defined. The place to define it is on page 5 (near the middle of the page) were ‘short-range stiffness’ is first mentioned.

4) Page 7. My previous review suggested that the Kandel reference (now reference [53]) should not be used for the explanations included on page 7 since this is a general textbook where it would be very hard to find what is discussed in this section. The original reference for the data shown in Kandel is the Peterka 2002 reference (now reference [36]). I also suggested that Figure 1A should include the proprioceptive weight measures shown for vestibular loss (VL) subjects. These VL proprioceptive weights are shown in Figure 10 of [36]. The authors apparently did not understand that Figure 10 does show proprioceptive weight measures from four VL subjects obtained in an eyes closed, surface-tilt stimulus condition that is relevant to the discussion in this section (as well as other conditions that are not relevant here). Another reference that more clearly presents that Peterka 2002 weights for the eyes closed, surface tilt condition is in Figure 2.5A of the reference:

Peterka, R.J., Sensory integration for human balance control. Handbook of Clinical Neurology, Vol 159 (3rd series) Balance, Gait and Falls, B.L. Day and S.R. Lord editors, 159:27-42, 2018.

5) Page 9 near the middle of the page, the sentence starting ‘Similar to the experiments, . . ‘. Suggest to use ‘ . . . for the obstacle task compared to the circle task as the second part . . .’

6) Page 17, equation (11). This equation included the variable ‘a’ but the next line refers to ‘a_base’.

7) Page 23 just before the [81] reference. Units should be (Nm)^2/Hz.

8) Page 5 of ‘Supplementary Information’ document. At the bottom of page 5 there are 5 equations shown but the equation numbers include equation (4) for which there is no actual equation shown. Assuming equation (4) should not exist, its elimination would affect subsequent equation labeling.

9) Page 6 of ‘Supplementary Information’ document. Should the gamma variable in equation (10) include a ‘i’ subscript as in equation (8) in the main manuscript?

10) Page 8 of ‘Supplementary Information’ document. Between equations (27) and (28) there is the ‘s.t.’ abbreviation. Better to spell out ‘subject to’.

11) Page 10 of ‘Supplementary Information’ document. Under point (2) of the ‘We observed the following problems.’ section, better to say ‘. . . must be chosen to be smaller, but . . .’

Reviewer #3: I am entirely satisfied with the author's responses and manuscript revisions.

**Have the authors made all data and (if applicable) computational code underlying the findings in their manuscript fully available?**

Reviewer #1: Yes

Reviewer #2: Yes

Reviewer #3: Yes

PLOS authors have the option to publish the peer review history of their article (what does this mean?). If published, this will include your full peer review and any attached files.

Reviewer #1: **Yes: **Bastien Berret

Reviewer #2: No

Reviewer #3: **Yes: **Antonie J. van den Bogert

Figure Files:

Data Requirements:

Reproducibility:

References:

---

## [Editor Report · Decision Letter 2]

11 Apr 2022

Dear Mr Van Wouwe,

We are pleased to inform you that your manuscript 'An approximate stochastic optimal control framework to simulate nonlinear neuromusculoskeletal models in the presence of noise' has been provisionally accepted for publication in PLOS Computational Biology.

Best regards,

Adrian M Haith

Associate Editor

PLOS Computational Biology

Wolfgang Einhäuser

Deputy Editor

PLOS Computational Biology

---

## [Editor Report · Acceptance letter]

9 May 2022

PCOMPBIOL-D-21-01394R2 

An approximate stochastic optimal control framework to simulate nonlinear neuromusculoskeletal models in the presence of noise

Dear Dr Van Wouwe,

I am pleased to inform you that your manuscript has been formally accepted for publication in PLOS Computational Biology. Your manuscript is now with our production department and you will be notified of the publication date in due course.

With kind regards,

Anita Estes
